# Insights into unique anatomical structures of the ascidian *Halocynthia papillosa* obtained by multimodal imaging

Lukas Hessel[1,2], Jonas Albers [1], Annika Michalek[3], Til Böttner [2], Elizabeth Duke[1], Ida Siveke[2], Stefan Herlitze[2], Jürgen Goldschmidt [3] & Mareike Huhn [2] ✉

Current understanding of anatomical structures of ascidians remains limited. This study presents multimodal imaging techniques, including Light, Thunder, and fluorescent confocal microscopy, to investigate neural structures and the tunic of *Halocynthia papillosa*, a common ascidian in the Mediterranean Sea. We demonstrate advanced 3D imaging methods, i.e., Magnetic Resonance Imaging, and High-Throughput Tomography (HiTT) at a synchrotron beamline. Imaging results show structural differences in the central nerve of *H. papillosa* compared to other ascidians and identify three distinct suborders of oral tentacles. We also document detailed autofluorescent patterns in ascidian cuticular sheds for the first time. HiTT imaging of the tunic reveals a spiralized structure emerging from cellulose layers. The state-of-the-art imaging techniques presented here encourage a broader use of HiTT to study functional anatomy in marine invertebrates. It establishes a strong foundation for future studies on solitary ascidians and highlights the need to expand research beyond model species.

The ascidian *Halocynthia papillosa* (Linnaeus, 1767) is a solitary and sessile benthic filter feeder found in the Mediterranean Sea and along the Portuguese coast of the Northeast Atlantic[1]. Ascidians have become significant model organisms in recent years, primarily due to phylogenetic analyses that identify them as the closest relatives of vertebrates, forming a sister group within chordates[2,3]. Their classification as an evolutionary link between chordates and invertebrates has made them valuable subjects for various biological studies[4]. Ascidians are particularly interesting for ecological and developmental research, as well as in pharmaceutical studies and drug discovery. The two siphons – in the contracted state of the animal –, and the tunic serve as a physical barrier[5]. Numerous bioactive compounds and symbiotic interactions with bacteria in the ascidians' hemolymph and digestive tract contribute to an effective immune system, which is of particular interest to the development of human medicine due to their close phylogenetic proximity[6–8]. Ascidians play a crucial role in benthic ecosystems, particularly in the vertical transport of organic material[9]. Following filtration, ascidians assimilate essential nutrients while expelling waste as excretory pellets through the atrial siphon. These pellets subsequently settle on the seafloor, serving as a food source for scavengers and deposit feeders[10]. Therefore, ascidians, which significantly contribute to reef biomass, facilitate the aggregation of suspended organic matter, thereby enriching nutrient cycles and supporting the stability of the reef ecosystem. Their role in these processes highlights their ecological importance in marine environments.

Despite its high abundance in the Mediterranean Sea and the accessibility of its habitat in shallow coastal waters, *H. papillosa* has not been widely adopted as an experimental model organism. However, research on the anatomy and biology of this species was conducted from the early 1970s through the 2000s[11–15]. These studies primarily focused on histochemical and microscopic examinations of the tunic. In contrast, extensively studied ascidian model organisms, such as *Ciona intestinalis*, have been the subject of numerous investigations that cover nearly all physiological functions, including the nervous system and embryonic development[16–19]. Comparative analyses of the tunic across different ascidian species and detailed examinations of their neural structures reveal that while the fundamental anatomical features are conserved, they show significant variations in morphology and developmental patterns[20].

The tunic of ascidians is an extracellular matrix that serves protective, structural, and physiological functions[21]. It consists of tunicin, which is primarily composed of cellulose embedded within a matrix of proteins, glycoproteins, and sulfated polysaccharides[22,23]. The synthesis of cellulose is unique to the urochordates, which include the Ascidiacea, Thaliacea, and Appendicularia[24,25]. Cellulose in ascidians is synthesized by the CesA enzyme expressed in the outer cell membrane of epidermal cells[26,27]. The

[1]European Molecular Biology Laboratory, Hamburg Unit c/o DESY, Hamburg, Germany. [2]General Zoology and Neurobiology, Ruhr University Bochum, Bochum, Germany. [3]Combinatorial Neuroimaging Core Facility, Leibniz Institute for Neurobiology, Magdeburg, Germany.
✉e-mail: mareike.huhn@ruhr-uni-bochum.de

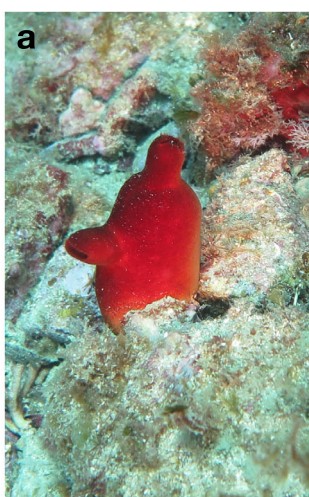
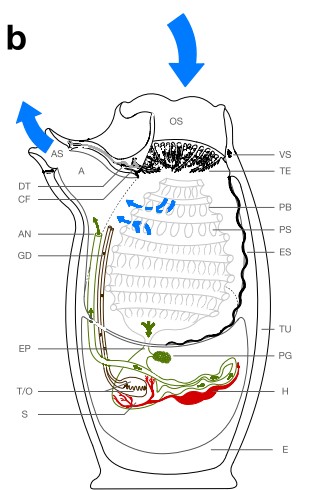
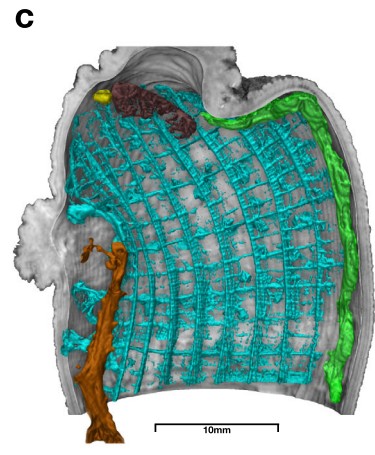

**Fig. 1 | General anatomy of *Halocynthia papillosa*. a** In situ picture of *H. papillosa* taken during scuba diving in Pula, Croatia, June 2022. **b** Scheme of *H. papillosa*, including water inflow (blue arrow, top), water outflow (blue arrow, left), genital flow (small brown arrows), and digestive flow (green arrows). The drawing shows a cross-section of the sagittal plane with the atrial siphon pointing to the left. **c** Three-dimensional rendering of the acquired MRI data (raw data shown in Fig. 2). Gray shows the tunic, green shows the endostyle, dark red shows the tentacle ring with an incision to free the view on the tentacles, yellow shows the dorsal tubercle, brown shows the compromise of digestive and genital tubes, and turquoise shows the pharyngeal basket. The figure shows a perspective illustration created with VG Studio Max, which, for the scale, is just an approximation. Abbreviations can be found in the following. A atrium, AS atrial siphon, OS oral siphon, DT dorsal tubercle, CF ciliated funnel, AN anus, GD genital ducts, EP esophagus, T/O testis/ovary, S stomach, E epithelium, H heart, PG pyloric gland, TU tunic, ES endostyle, PS pharyngeal slit, PB pharyngeal basket, TE tentacles, VS velar sphincter.

ascidian tunic is a living tissue that contains morula cells and bacteriocytes, contributing to the previously explained immune defense[21]. As this study will not particularly focus on the composition of the tunic matrix, the term "cellulose" will be used in the following sections.

Comprehensive microscopic analyses have characterized the structural organization of ascidian tunics, identifying distinct layers. The outermost region, known as the outer cuticle, contains species-specific scales, while the inner layer, called the fundamental layer, consists of the primary structural tissue[11]. Molecular studies have identified the main components of the *H. papillosa* tunic as sulfated acids, mucopolysaccharides, cellulose, and proteins[12]. The fundamental layer can be divided into distinct sub-layers. While most research has focused on early developmental stages, investigations have been limited to individuals up to 30 days post-morphogenesis, with no comprehensive assessments of adult specimens[13–15].

Even less is known about the neural system in *H. papillosa;* however, it has been well-studied in other ascidian species[20]. Originating from the tentacles, the subcoronal nerve (SCN) extends into the velar sphincter (VS), which encircles the entire lumen of the oral siphon. Nerves extending from the VS into the tentacles form a plexus where motor and sensory components are integrated and indistinguishable. Only a few afferents, with nuclei located in the central ganglion (CG), are distinguishable[28]. The ascending nerves of the coronal organ connect directly to the CG via a nerve bundle, forming a reflex loop with the velar sphincter and squirt muscles[29]. A study by Braun and Stach (2019) provides a comprehensive reference for an essential overview of the CG[20]. Their investigation, which examined 18 ascidian species using microscopic data, enabled the reconstruction of three-dimensional models, revealing variations in the number and organization of brain nerves. However, the study did not include any *Halocynthia* species.

The present study aimed to investigate the structure and composition of the tunic, as well as the structure and location of the oral tentacles, cerebral ganglion, and dorsal tubercle, in *H. papillosa* (Fig. 1). To thoroughly analyze tissue diversity, we employed a combination of microscopic techniques, molecular approaches, and high-throughput tomography X-ray imaging (HiTT) for three-dimensional reconstructions. Initial examinations used low-magnification microscopy, including light microscopy and Thunder microscopy, to provide an overview of neuronal structures and selected body regions, create true-color representations, and visualize auto-fluorescence in vivo. Subsequently, high-resolution imaging techniques, such as confocal microscopy, were employed to investigate fine structural details further. Autofluorescence imaging was used to generate three-dimensional maximum-intensity projections (MIPs) of thick tunic sections. In addition to the molecular characterization of tunic components using Fourier-transform infrared spectroscopy (FTIR), high-resolution three-dimensional imaging was conducted using the advanced HiTT technology from EMBL at DESY in Hamburg. The integration of these multimodal imaging methods offers a detailed and multidimensional perspective on neuronal structures and tunic architecture, providing insights into their organization and composition.

## Results
Imaging techniques at various scales were utilized to enhance understanding of the anatomical structures of the solitary ascidian *Halocynthia papillosa*. The imaging results presented here range from the level of the whole animal (3 × 6 cm) to the details of neural structures (2–100 μm).

### Whole animal investigations
Magnetic resonance imaging (MRI) data was collected from a single individual ascidian. To achieve a comprehensive and detailed visualization of internal structures, high-resolution 2D images (100 × 100 μm in-plane) were acquired in both transverse (Fig. 2a, b) and sagittal (Fig. 2c, d) orientations. Additionally, isotropic imaging was conducted at a lower resolution (250 × 250 x 250 μm) for three-dimensional renderings of prominent structures (Fig. 1 c, Supplementary Movie 1). The strong contrast in the T2-weighted images allowed for a clear differentiation of internal structures between the internal body and the surrounding tunic. Within the internal body, the pharyngeal basket (PB) was visible in both the sagittal and transverse images (Fig. 2b, c). In the transverse view, the arrangement of PB structures at a 90° angle became clear. The endostyle extending from posterior to anterior was highly distinguishable against its surroundings (Fig. 2a–c). Furthermore, the stomach and intestine, with the latter extending to the atrial siphon, were identified (Fig. 2d). With additional contrast adjustments, neural structures like the dorsal tubercle (DT) also

Fig. 2 | Overview of MRI data. T2-weighted MRI images of an ascidian submerged in Fomblin® in transversal (**a**, **b**) and sagittal slices (**c**, **d**) with an in-plane resolution of 100 μm. The number of the displayed slice is given in the lower left of each image, orientation of the ascidian within the images is shown in the lower right. Red dotted lines in (**a** and **c**) show the corresponding orthogonal section planes (**a'**–**d'**). In the top transversal slice (**a**), a dotted circle marks the OS; the DT and ES are marked by arrows. Measured DT dimensions are included in green (width) and blue (length). In a more posterior section (**b**), the ES, the PBL, the M, and the AS are visible and marked accordingly. Black arrowheads outline the structure of the PBL throughout the images **b**–**d**, while white arrowheads show the tunic (**b**, **c**). In the sagittal midline slice of the ascidian (**c**), the ES, OS, PBL, and AS are displayed. **d** Right lateral side of the ascidian with marked M, INT, and S. OS oral siphon, DT dorsal tubercle, ES endostyle, PBL pharyngeal basket lumen, M mantle, AS atrial siphon, INT intestine, PB pharyngeal basket, S stomach, a anterior, p posterior, d distal, v ventral, rl right lateral, ll left lateral.

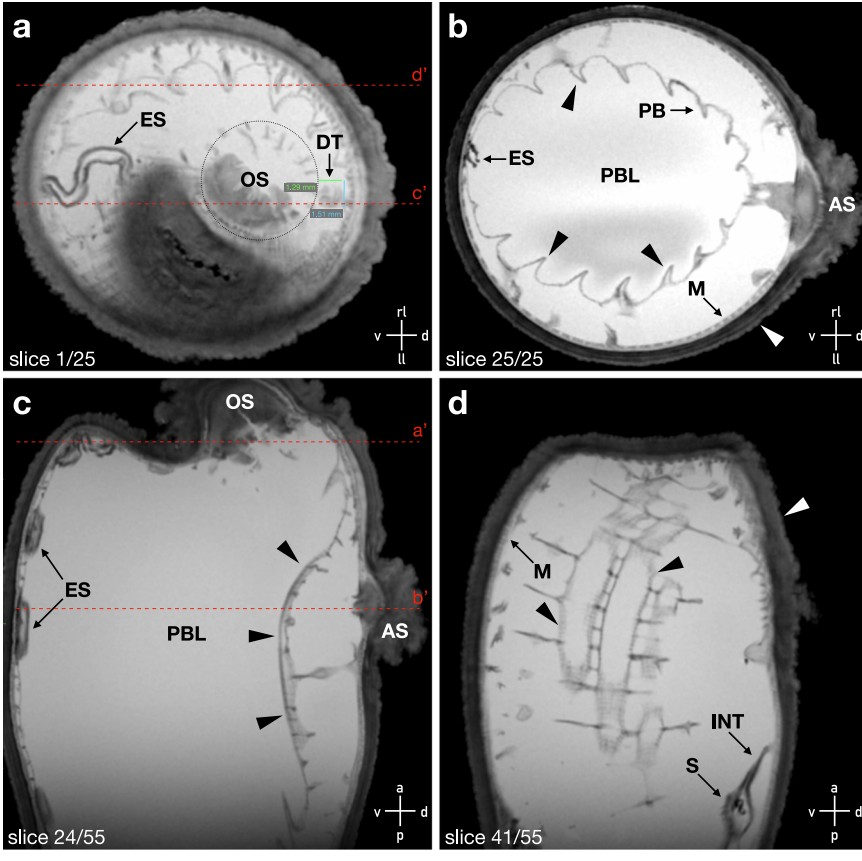

became apparent, allowing for measurements of its dimensions. The DT measured 1.51 mm in width and 1.29 mm in length as determined from the transverse images (Fig. 2a).

## Tunic investigations

For a general overview of the tunic, we imaged slices from five different individual ascidians using light microscopy. Light microscopy revealed the tunic structure in the cuticular zone (CZ), which contains the cuticular shed (CS) and the underlying fiber bundles (Fig. 3b), the median zone (MZ) that houses numerous cellulose layers with varying densities of embedded color pigments (Fig. 3b, c), the peri-epidermal zone (PE) characterized by a high density of nuclei and vacuolar cells[11], and the epithelium (E), a thin layer that separates the internal body from the tunic itself (rose line in Fig. 3a). The subcuticular zone (SC) could not be visualized using the magnification applied with light microscopy and is therefore only presented schematically (Fig. 3a). The integration of various imaging techniques (x-ray, light microscopy, and DAPI staining) enabled the creation of a comprehensive scheme that summarizes all the information (Fig. 3a)[11,12]. The appearance of the CS and MZ varied among samples from different individuals and across various sections of the same individual. In tunic sections that had not been exposed to sunlight (left or right side of the animal, depending on its orientation in the reef, and animals collected from caves or beneath overhangs), the MZ appeared pale white with barely distinguishable cellulose layers (Fig. 3b), while in sections with intense light exposure, the MZ displayed an orange to deep red hue (Fig. 3c). Overall, the MZ exhibited a color gradient, ranging from pale (on the body side not exposed to light) to dark (on the body side exposed to light). When the red color pigments were visible in the cellulose layers, a gradient (decreasing from outside to inside across cellulose layer groups) and a sub-gradient (increasing from outside to inside within each cellulose layer group) were observed (Fig. 3c).

To investigate potential differences in tunic spectral reflectance between contracted and relaxed animals, spectral intensity maxima were identified for each combination of condition (contracted vs relaxed) and body region (dark and pale body side). For each of the four combinations, the wavelength with the highest mean intensity was determined based on averaged curves across 13 animals (Fig. 4a, Supplementary Data 1). At these peak wavelengths, individual intensity values were extracted for each animal (Fig. 4b, Supplementary Data 2). To examine the overall effects, a two-way repeated-measures ANOVA was performed to assess the main effects of condition and sampling point, as well as their interaction. The analysis revealed significant main effects of contraction ($F(1,11) = 195.71$, $p < 0.001$) and body region ($F(1,11) = 29.29$, $p < 0.001$), with F indicating the ratio of explained to unexplained variation. The interaction effect was not significant ($F(1,11) = 0.53$, $p = 0.483$). A one-way ANOVA with repeated measures (animal ID) and Bonferroni corrected p-values was then conducted with keeping the sampling point (body side) constant to assess the contraction effect. On both sides, mean intensity was higher in the contracted state (dark body side: $F(11) = 79.7$, $p(\text{adjust}) < 0.001$; pale body side: $F(11) = 89.4$, $p < 0.001$), with F indicating the ratio of explained to unexplained variation. Effect sizes (Cohen's d) were large for both body regions (dark side: $d = 4.15$; pale side: $d = 2.73$).

The CS appeared more consistent among slices from different individuals and various slices of the same individual in terms of color and overall shape. The scales were light orange, regardless of the animal's location and coloration (Fig. 3b, c). Higher-magnification detailed images of the tunic, taken with a Thunder microscope in vivo (top view) and a Confocal microscope (in slice), showed strong autofluorescence exclusively in the CS (Fig. 3d–l), but not in the underlying tissues (Fig. 3f, i). The fluorescent pattern of the CS allowed for comparisons between the tunic of relaxed individuals (Fig. 3d–f) and contracted individuals (Fig. 3g–i). At low magnification, dark, non-fluorescent spaces between the CS were evident in relaxed individuals (Fig. 3d). In contracted individuals, no spaces were observed, and the fluorescence covered the entire tunic surface (Fig. 3g). At

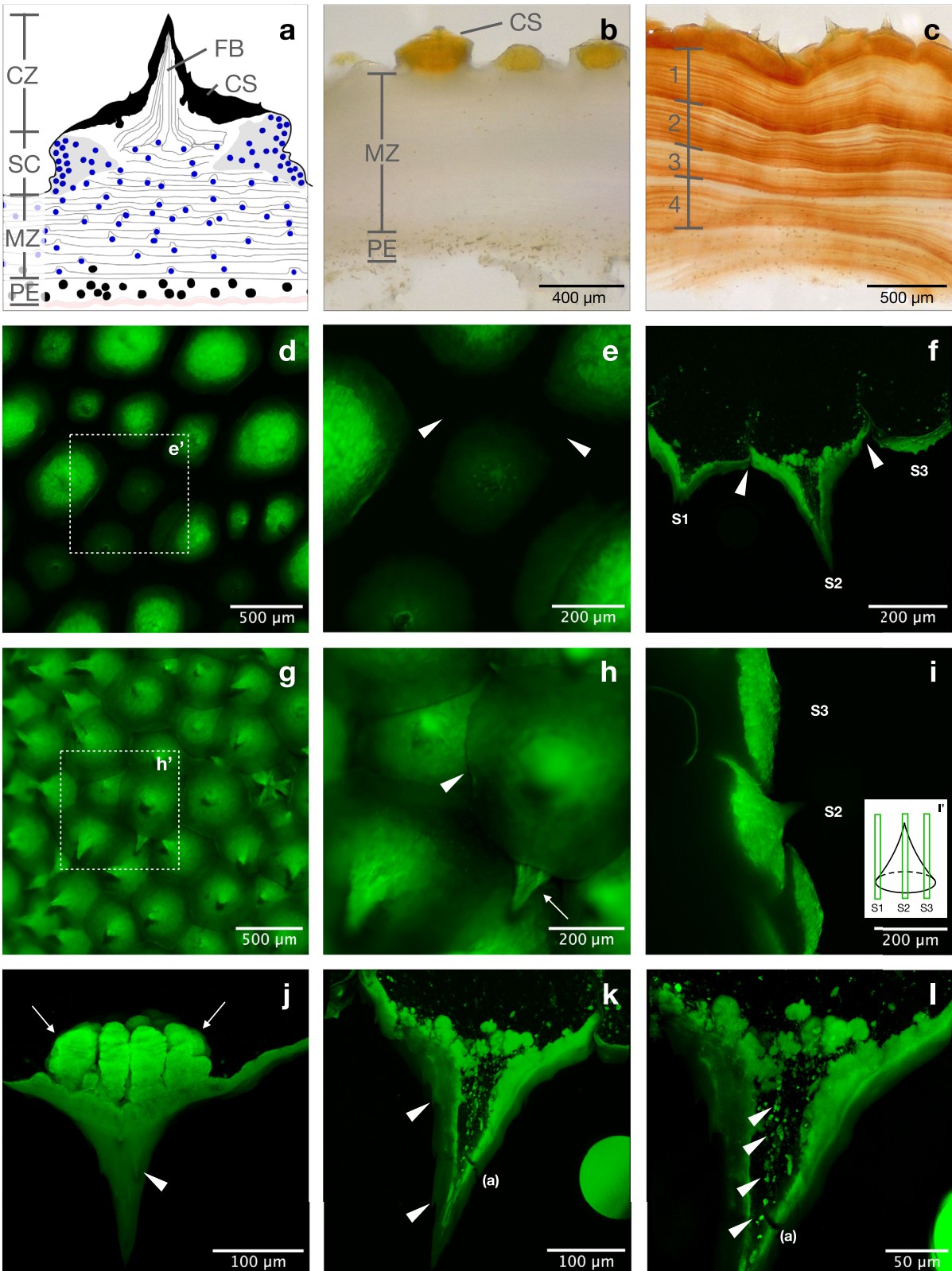

higher magnification, a superposition of neighboring CS became clear (Fig. 3h).

The superposition of different CS was not uniform, with some neighboring CS overlapping more than others (white arrow in Fig. 3h). This variation was also visible in the slices (Fig. 3f, i). The different imaging angles and magnifications revealed a superposition of the CS during contraction, with

some CS being subordinate to others. Various sectional planes of the CS could be depicted (Fig. 3f, i) and illustrated in a scheme (Fig. 3i'). Higher magnification images of the CS structure were obtained through confocal microscopy (Fig. 3j–l). The CS structure appeared conical and tapered, with small tips aligned with the main axial tip on the surface (white arrowheads, Fig. 3j, k). In the medial direction, rounded, bubble-like structures beneath the CS were

**Fig. 3 | Tunic imaged with different microscopes at different magnifications.** Tunic slices displayed as scheme (**a**), light microscopic images (**b**, **c**), and Thunder microscopic images (**d**, **e**, **g**, **h**, **i**) and Confocal imaging (**f**, **j**–**l**) in different magnifications. The scheme (**a**) comprises all features displayed in Fig. 3 (**b**–**l**) and Fig. 5. **b** Thin black lines indicate the cellulose layers, while blue dots represent DAPI-stained cell bodies in the SC and MZ, whereas black dots show cell bodies in the PE. Thick tunic section from an individual located in a sunlight-protected environment in a relaxed state. **c** Individual from a location with regular light exposure in a more contracted state. The numbers (Fig. 3c) indicate the four cellulose layer groups with decreasing intensity of color (outside to inside, with increasing numbers). The increasing intensity and color (from outside to inside) are visible within each group. **d** Low magnification of the autofluorescent CS in a not-contracted (relaxed) state with non-fluorescent spaces in between. **e** Same tunic view as (**d**) in a higher magnification at location (**e'**) with white arrowheads pointing to the non-autofluorescent spaces between the CS. **f** Maximum intensity projection (MIP) of a thin tunic section

acquired with a confocal microscope, visualizing three different cross-sections of a CS (S1-S3) as indicated in (**i'**). Surface pointing to the bottom. **g** Tunic section similar to (**d**, **e**) in a contracted state. **h** Same tunic as (**g**) in a higher magnification at position (**h'**) with a white arrowhead pointing at the overlap of two CS and a white arrow pointing at the submerged Spine of a CS. **i** Contracted tunic cross-section imaged with a Thunder microscope, visualizing the superimpositions of the CS. Surface pointing to the right. Annotations S2 and S3 show the respective cross-sections illustrated in (**i'**). **j**, **k** High magnification (40x oil lens) MIP of a thin tunic section (50 μm) with white arrowheads pointing to small sub-spines on the central spine. Surface pointing to the bottom. White arrows show the bubble-like structure below the spine (**j**), while (**a**) shows an artifact not to be misinterpreted. **l** Higher magnification (63x oil lens) of (**k**) with white arrowheads pointing to rounded particles lined up like a string. Surface pointing to the bottom. (a) artifact, CS cuticular shed, CZ cuticular zone, FB fiber bundle, MZ median zone, PE peri-epidermal zone, SC subcuticular zone.

**Fig. 4 | Spectral intensity measurement.** Spectral intensity curves on the red and light side of the body of *Halocynthia papillosa* in the contracted and relaxed state. **a** The average spectral intensity of 12 individuals is shown as a function of wavelength (400–900 nm) for two body regions (red and light side) in two physiological states (contracted vs. relaxed). Solid lines represent the contracted state, while dashed lines represent the relaxed state. The red side of the body is shown in dark red, and the light side is shown in light brown. Shaded areas around each line show the 95% confidence intervals. Mean values were calculated for 12 animals per condition. **b** Spectral intensity maxima (means ±95% confidence intervals, $n = 12$) measured on the red and light side and in the contracted and relaxed state. Colored circles (red for measurements on the red body side and beige for measurements on the lighter body side) show the data distribution within each factor combination (body side and contraction status (relaxed and contracted)).

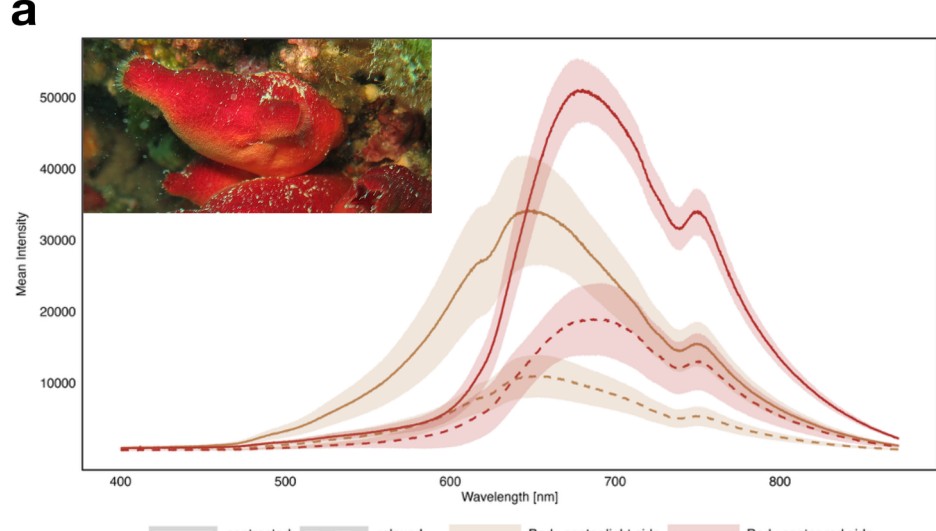

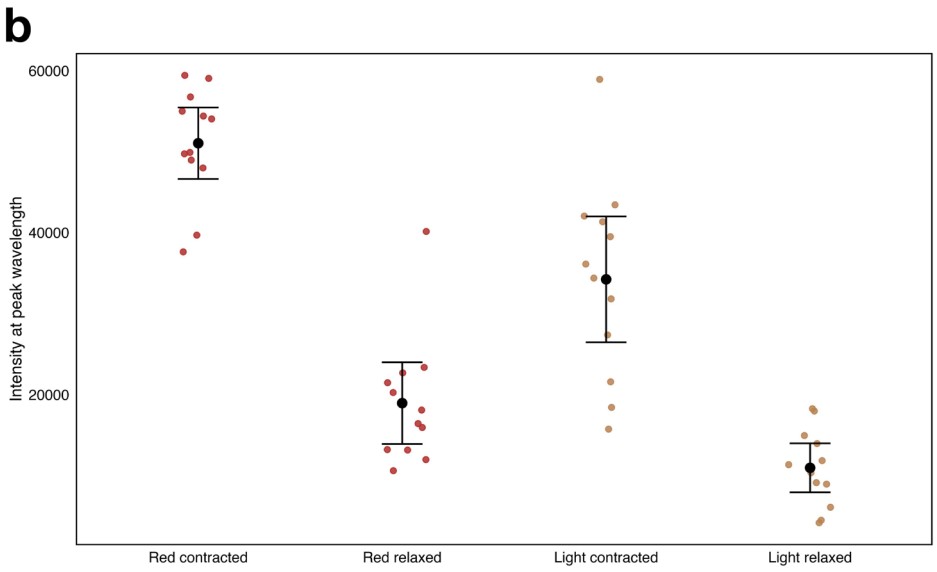

visible, oriented in alignment with the overall orientation of the cone (white arrows, Fig. 3j). The subcuticular structure and fluorescence resembled the outer surface of the cone. A channel was visible, narrowing towards the tip. Higher magnification (63x) revealed fluorescent particles arranged like a string toward the tip (white arrowheads, Fig. 3l). The presence of fiber bundles had

been established in earlier investigations, with the nuclei situated below the spine[12]. DAPI staining highlighted the cell nuclei and their distribution in the cellulose layers and around the CS. Nuclei of different cell types were absent in the CS and within the CS channel (Fig. 5a). They were regularly distributed in the cellulose layers and occurred in greater density around the CS.

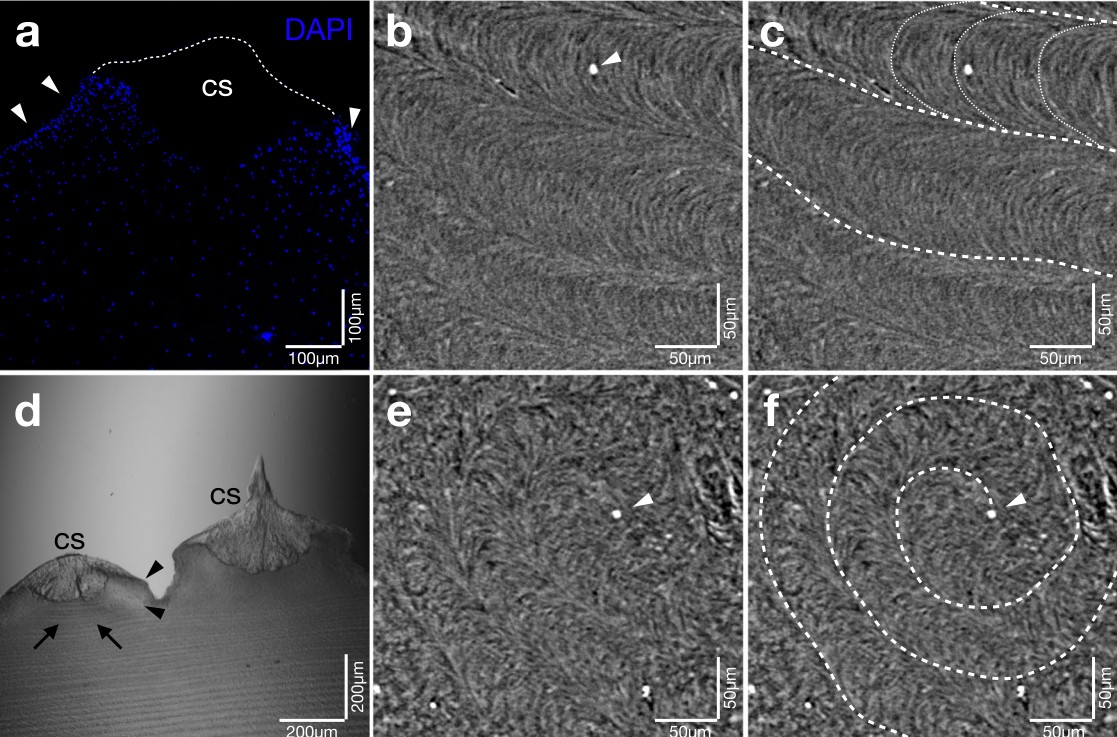

**Fig. 5 | Cellulose fine structure. a** DAPI-stained slice of the tunic acquired with a Thunder microscope. The dotted line shows the CS that is not visible in the DAPI staining. White arrowheads show a higher density of stained nuclei at the position of the spiralized cellulose. **b** Cellulose structure in the top view within the MZ acquired with HiTT, with a white arrowhead pointing at a fiber bundle in the cross-section. **c** Same section of cellulose as (**b**) with additional annotations to visualize the parallel aligned cellulose fibers (coarse dotted line) and the crescent-shaped fine structure of the cellulose fibers within a bundle (fine dotted line). **d** Part of a cross-section of the tunic of *H. papillosa*, imaged with a Thunder microscope in transmitted light to visualize the parallel cellulose fiber layers in the middle of the CS (black arrows), which spiral towards the CS (black arrowheads). Additionally, two different cross-sections of the CS itself are visible. **e, f** Same section of cellulose directly below a spine in a cross-section from the top view. The white arrowhead shows the fiber bundle in the center of the rising spiral. The dotted white line indicates the spiralized cellulose structure, characterized by a crescent-shaped fine structure similar to the parallel regions (**b**–**d**). AS, atrial siphon; CS, cuticular shed; OS, oral siphon.

For the structural tunic analysis of the three-dimensional HiTT X-ray images, we selected a sectional plane in the middle of the tunic, approximately equidistant from the epithelial tissue and the spines. In this top view of the tunic, a parallel arrangement of lanes, 50-150 µm wide, was visible (Fig. 5b, c). These lanes exhibited a crescent-shaped structure and spiraled in some areas. Neighboring lanes were aligned in the same direction, occasionally interspersed with white (i.e., very X-ray-dense) structures that appeared as tiny dots in the sectional image (white arrowhead, Fig. 5b). The spiralization observed in the top view (Fig. 5e, f) and the side views of similar positions (Fig. 5d) suggests that the cellulose forms a cone-shaped depression (Fig. 5a) from which the fiber bundles (white arrowheads, Fig. 5b, e) and the CS emerge. The three-dimensional HiTT images supported this assumption. Even within the spiralized cellulose, the crescent-shaped structures remained intact. The dense, white dot-like structures, presumed to be the fiber bundles, were located precisely in the center of the spiralized cellulose (Fig. 5e, f). In the side view, the less complex layered structure of the cellulose was visible (arrows, Fig. 5d). This simple layered structure faded at the elevation of the CS (arrowheads, Fig. 5d).

In addition to the anatomical and structural investigations, Fourier transform infrared spectroscopy (FTIR) was conducted on the tunic slices. The analysis included 10 measurement points evenly distributed over a 10 µm-thick tunic slice. Measurement points 1–8 were located in the MZ of the tunic, measurement point 9 in the PE, and measurement point 0 in the SC and CZ (Fig. 6). Absorbance peaks were found at 1060 cm$^{-1}$, 1035 cm$^{-1}$, 1336/7 cm$^{-1}$, 1430 cm$^{-1}$, and 985 cm$^{-1}$, and were identified as cellulose-like or tunicin complex[30]. A protein compound was detected between 1500 cm$^{-1}$ and 1700 cm$^{-1}$ (absorbance peak at 1547 cm$^{-1}$ corresponding to amide band II absorption, and 1650 cm$^{-1}$ corresponding to amide I absorption). The two other prominent peaks around 2800 –3000 cm$^{-1}$ can be assigned to the C-H stretching bonds, while the second most prominent peak complex (3200 –3500 cm$^{-1}$) exhibited numerous OH-bonds[30].

## Nervous system investigations

The cerebral ganglion (red box, Fig. 7a) was identified as a continuous cord that bifurcates dichotomously twice before each siphon (white arrowhead and black outlined arrowhead, Fig. 7b) and encircles the siphon like a ring. From this surrounding ring, individual strands branched off multiple times, innervating the individual tentacles. A row of adjacent muscle strands was arranged parallel to the nerve (Fig. 7c, d, darker blue), while an outer layer of muscle strands was found perpendicular to both the nerve and the inner strands (lighter blue). These muscle strands formed bundles (Fig. 7c, d), which were also visible in the light microscope images (black arrowheads, Fig. 7b). Using a stitching protocol developed by EMBL Hamburg[2], the entire dataset (total length 8492.25 µm) acquired by HiTT in various orientations could be visualized in a three-dimensional reconstruction (Fig. 7e, f). The dorsal strand plexus (DSP) could also be visualized and was found to be partially running parallel to the nerve wrapping around it until it lost contact with the nerve and disappeared from the field of view before the nerves divided dichotomously near the atrial siphon (Fig. 7e, f). Towards the oral siphon, the DSP appeared to be covered by muscles (Fig. 7e). At this point, the DSP seemed to emerge from the nerve, and together they appeared to form cavities (black arrowheads, Fig. 7g). The nerve did not exhibit any apparent thickening or structural changes at any site that could be identified as the CG (Supplementary Movie 2).

The dorsal tubercle (DT) of *H. papillosa* was examined using various imaging techniques (Fig. 8). It was located above the nerve cord, just anterior

**Fig. 6 | FT-IR measurement of the tunic thin section.** The entire spectrum of the FT-IR measurement was performed on a thin section of the tunic. The slice in the upper center of the figure shows the thin tunic section with its exact measurement points shown as colored dots in the magnified section (box on the right). The orientation is from outside (left) to inside (right), and measurement points are numbered from outside to inside. The Diagram shows the Wavenumber (cm−1) on the x-axis and the absorbance units on the y-axis. The graphs are numbered according to the measurement position indicated at the top.

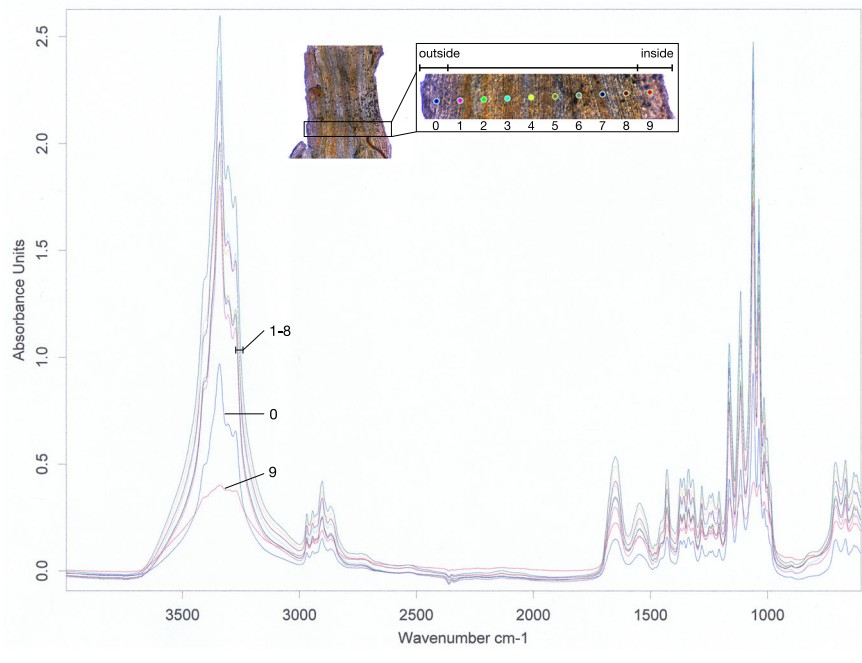

to the oral siphon (OS), and became visible under a light microscope following dissection (Fig. 8b). In the low-magnification image, both siphons (OS and AS, Fig. 8b) and the entire central nerve cord (black arrowheads, Fig. 8b), including the initial dichotomous branching of the nerve near the atrial siphon, were identifiable (white arrowheads, Fig. 8b). Adjacent to the oral siphon, the DT obscured the branching of the central nerve in the light microscopic view. The DT had a diameter ranging from 2 to 5 mm (depending on the animal and measurement direction) and featured a ciliated funnel that appears horseshoe-shaped with elevated spirals within a jelly-like shell (Fig. 8h, i), as observed and investigated previously in *Microcosmus bitunicatus* and *Halocynthia roretzi*[20,31]. The elevation of the horns became apparent in the lateral view (Fig. 8g, h). Light microscopy displayed the transparent tissue along with a yellow-to-orange hue of the ciliated funnel (black arrowheads, Fig. 8i). The HiTT acquisitions of the DT were conducted on tissue that underwent a dehydration series with ethanol to enhance contrast during scanning. Consequently, the DT appeared shrunken in the HiTT images (Fig. 8c–f). Three-dimensional rendering visualized the DT's isolation from the surrounding tissue (Fig. 8d), the rear view free of surrounding tissue (Fig. 8e), and a segmentation of the central nerve cord beneath the DT (Fig. 8e). This segmentation revealed the dichotomous nerve division located directly beneath the DT (Fig. 8e, f).

### Oral tentacle investigation

Another area of focus was the oral tentacles. The oral tentacles of *H. papillosa* were arranged in a ring within the oral siphon, oriented directly into the inflowing water stream through the siphon (OS, Fig. 9a). Each tentacle featured smaller sub-tentacles emerging from its primary structure (Fig. 9a', b–c). These sub-tentacles were attached to the lower (posterior) side of the main tentacles (Fig. 9b, c). The tentacles appeared rounder toward the outer side of the oral siphon (Fig. 9b) and flatter toward the inner side of the internal body (Fig. 9c). Generally, the tentacles tapered in the medial direction (Fig. 9b, c). Segmentation of the HiTT images revealed nervous structures (green) and blood vessels (red) within the tentacles (Fig. 9d–e). Blood vessels extended posteriorly inside the tentacle, branching into each sub-tentacle (Fig. 9c). When the surrounding tissue was removed in the 3-dimensional projection, the segmented blood vessels (red) and nerves (green) became visible (Fig. 9d, e). In partial isolation, where only half of the tissue was removed, additional details about the branching of nerves and blood vessels in the sub-tentacles became evident (black and black outlined arrowheads, Fig. 9d). Many smaller vessels were present inside the tentacles, but were not included in the visualization. For clarity, only the larger vessels (first and second order) were segmented and displayed (Supplementary Movie 3).

## Discussion

Using multi-scale imaging, we characterized the neural, oral tentacle, and tunic structure of *Halocynthia papillosa* in unprecedented detail. Our data reveals anatomical features that diverge from established patterns in solitary ascidians.

### Tunic investigations

The tunic of ascidians shows considerable variation across taxa. For example, in *Ciona intestinalis*, the tunic consists of a gelatinous and thin tissue[32], whereas in *Halocynthia* spp., it has a leatherier consistency[33,34]. Notably, even within the same genus, such as in *Halocynthia* spp., distinct differences in the tunic's structure have been observed[35]. Early studies on *H. papillosa*[12–15,36] described its robust tunic and distinctive cuticular sheds (CS). At the observation site, spines similar to those found close to the siphons in *H. roretzi* were found[7,37]. When the animal contracts, these cuticular sheds form a strong armor that conceals the softer underlying tissues.

Furthermore, our findings reveal the presence of autofluorescence in the tunic. This autofluorescence was highly concentrated in the CS, which obscured non-fluorescent tunic regions during contraction (Fig. 3g, h). Cavities within the ~300-µm spines contained bead-like fluorescent particles transported along central fiber bundles, similar to observations during embryonic tunic development[15].

This constitutes only the second report of fluorescence in adult ascidians[38]. Although fluorescence can be an incidental function[39], the functions of fluorescence in marine taxa remain poorly studied[40], and whether contraction-dependent changes in fluorescence serve functions such as camouflage remains unclear.

Spectral measurements of light-exposed and darker tunic regions corresponded with these fluorescence patterns. Contraction increased reflectance intensity, and spectral signatures differed consistently by body region. These effects suggest that muscular contraction may modulate the tunic's optical appearance, potentially contributing to ecological functions such as camouflage against red-dominated coralligenous backgrounds. Further investigation is needed to confirm this.

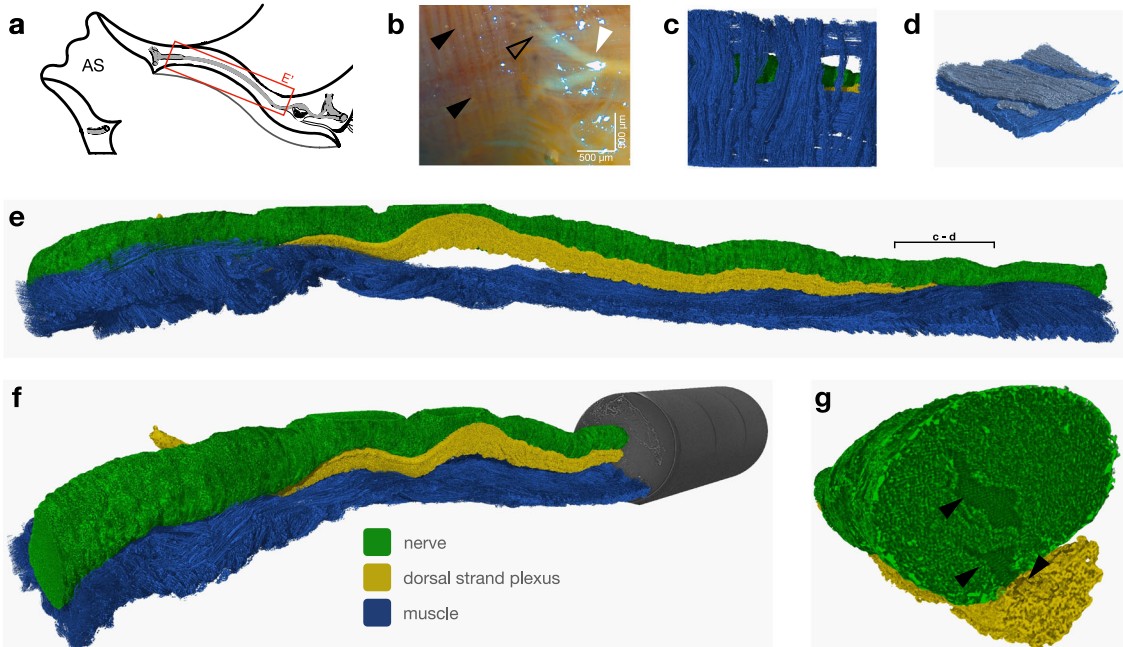

**Fig. 7 | Nerve, dorsal strand plexus, and muscles between the siphons. a** Schematic representation of the region investigated in the sea squirt. The red box shows the area from the dorsal tubercle (DT) on the right to the dichotomous split of the central nerve on the left. **b** Light microscopic image of the atrial siphon ring muscles (black arrowheads) and the first (white arrowhead) and second (black outlined arrowhead) dichotomous branching of the central nerve after resection of the tunic. **c–g** Three-dimensional reconstructed renderings made with VG Studio Max in a perspective view. Therefore, no scale is given. **c, d** Reconstructed and segmented subregion of the muscles in proximity to the nerve strand. Light blue muscle strands show the parallel to the central nerve-aligned muscle strands, while dark blue muscle shows the orthogonal muscle strands. The large bracket marked in (**e**) shows the subsection. **e** Stitch of the acquired HiTT data set of the nervous system between the two siphons with a total length of 8492.25 μm, which is equivalent to the red box (**E'**) in (**a**). **f** Different perspective of the same dataset with a visualization of the non-segmented whole field of view (FOV) in the right part, including a legend that applies to all subfigures (**c–g**). **g** Specific region where the DSP emerges from the nerve and builds cavities in both structures, which are pointed out by black arrowheads. AS atrial siphon.

A recent study on *H. roretzi* by Song et al. (2020)[35] revealed the orientation of a tunic section in the media zone using scanning electron microscopy. We were able to reproduce and validate this figure using the HiTT imaging in *H. papillosa* with the advantage of freely adjustable virtual cross-sections. HiTT imaging further revealed spiralized cellulose fibers within the subcuticular zone (SC), forming a support structure for the CS and accommodating fiber bundles running to spine tips. FTIR analysis corroborated the presence of cellulose in the tunic's central layers, consistent with findings in *H. roretzi* and *C. intestinalis*[35,41]. Variation between studies likely reflects differences in scanner settings, while the outer and inner tunic layers showed complex spectra due to mixed tissue types.

## Nervous system investigations

The central nerve of *H. papillosa*, for example, showed no visible thickening that would allow localization of the cerebral ganglion (CG). In the reconstructed CGs of 14 species published by Braun and Stach[20], all Stolidobranchia showed distinct thickening of the nerve at the CG. For the genus *Halocynthia*, only one study (on *H. roretzi)* published images of the CG, and neither found any thickening in the central nerve[42]. In the respective study, the authors refer to the entire nerve between the two ends of dichotomous branching as the CG[42] and report this section to be 10 mm long[42]. In the present study with *H. papillosa*, we measured the nerve section between the two dichotomous branches to be 65 mm long, far exceeding the known range for solitary ascidians[20]. We therefore suggest that the CG of *H. papillosa* comprises only a subsection of this nerve region and is not externally distinguishable. Sub-micrometer imaging using HiTT could not identify anatomical differentiation within the central nerve. Future work using stains such as Nissl, MAP2, BP102, or Pan-Nav1, as well as in vivo Ca2+ imaging, will be required. As in other Stolidobranchia[20], the CG may lie near the anterior dichotomous branching underneath the dorsal tubercle (DT). Repeated dichotomous branching - which is sparsely described for the

investigated species[43] - was observed in *H. papillosa*. These branches ultimately merged with the circular oral or atrial siphon nerves. Our data also revealed internal nerve cavities at the emergence point of the dorsal strand plexus (DST).

## Oral tentacle investigations

The precise and three-dimensional anatomy of oral tentacles, described previously in *Botryllus schlosseri*[44], *Corella inflata*[28], *Styela plicata*[45], and *Ciona robusta*[46], is here documented for *H. papillosa* for the first time. The tentacles of *H. papillosa* divide into three sub-orders (compared with four in *M. socialis*[47]) and are rounded superiorly, likely decreasing resistance to incoming water flow and enabling the coronal organ to form a continuous fringe along the inferior margin (Fig. 9b). Sub-tentacle organization within Stolidobranchia is, therefore, inconsistent[44], suggesting a possible secondary loss of sub-tentacles in some families. Broader taxon sampling is needed to clarify this pattern.

Segmentation of tentacle structures revealed a vascular network consistent with a closed or semi-closed system, supporting the interpretations by Konrad (2016)[48]. First-order capillaries measured 79.0/68.1 μm (height/width), decreasing to 33.3/34.0 μm in second-order branches, while third-order vessels were below measurable resolution. A single inferior intra-tentacular vessel supplied each sub-tentacle, with alternating left–right subdivision. A corresponding innervating structure on the superior side mirrored this branching pattern. Smaller capillaries supplied the nerve, whose diameter was comparable to that of first-order vessels. Future work relating tentacle anatomy to environmental factors such as particulate load or acoustic conditions may clarify functional adaptations.

Overall, our findings provide a comprehensive anatomical examination of the tunic, oral tentacles, and selected neural structures of *H. papillosa*. Such data are vital for understanding the ecological roles of ascidians, which

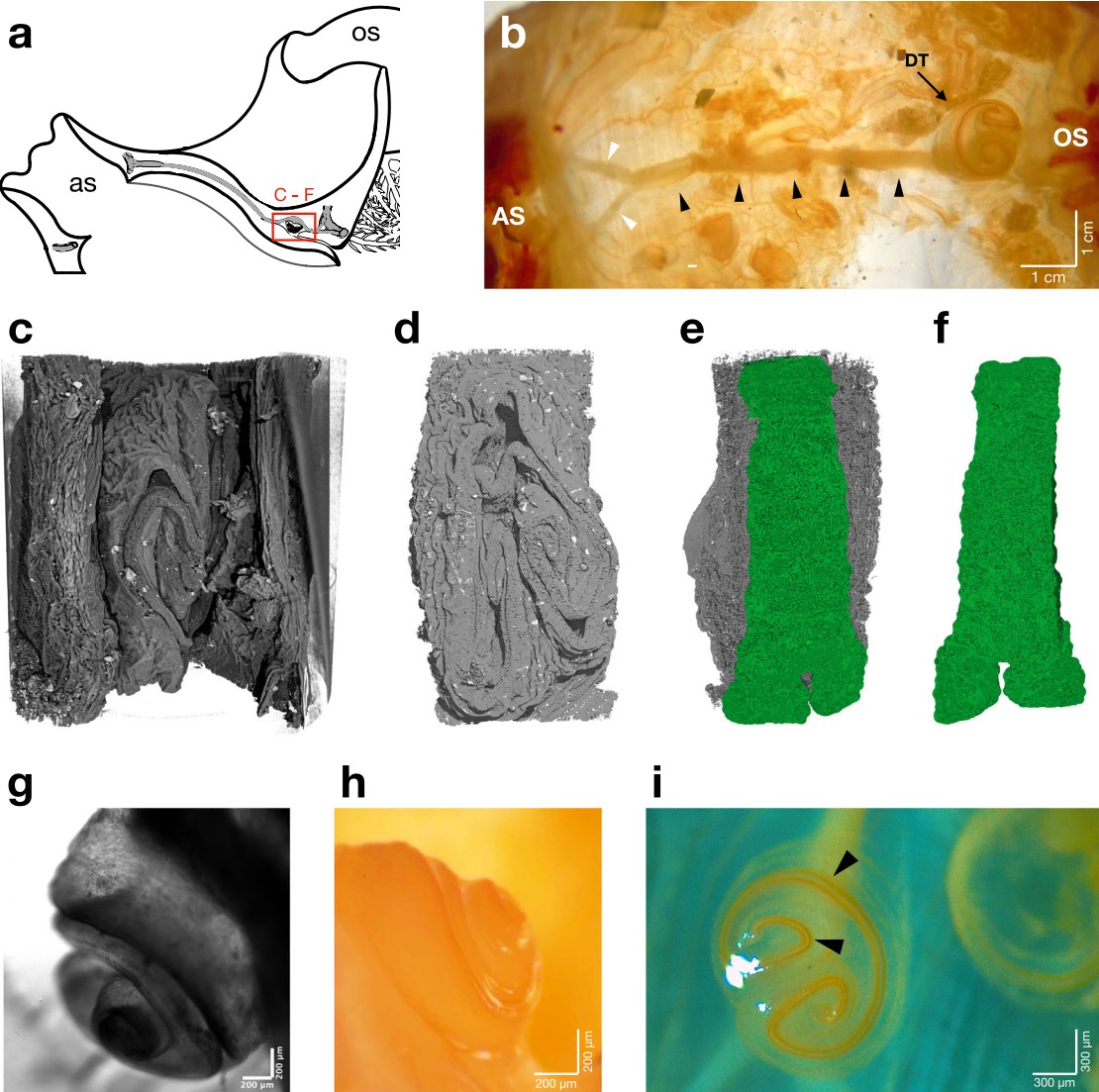

**Fig. 8 | Dorsal tubercle visualized by different techniques. a** Schematic drawing of the position of the dorsal tubercle and the central nerve. The dorsal tubercle is additionally marked with a red box. **b** Entire central nerve from the atrial siphon (AS – left) to the oral siphon (OS—right). The course of the nerve is shown with black arrowheads. The nerve splits dichotomously on the left (white arrowheads), resulting in the ring nerve surrounding the AS. The same happens on the right side (OS) but is covered by the dorsal tubercle (DT). **c** Three-dimensional rendering of the DT from the HiTT data. Due to the dehydration process, the structure shrank slightly. The same structure, without the surrounding tissue, is shown in (**d**). The position of (**c–f**) is shown by the red box in (**a**). Panels **e**, **d** with a longitudinal rotation angle of 180°. The green color shows the segmented nerve below the DT. **f** Segmented nerve separated with the dichotomous split visible. **g** Side view of the DT acquired with transmitted light from the Thunder microscope. **h**, **i** Light microscopic pictures demonstrating the structure's color and texture. Black arrowheads point at the ciliated funnel. Figures (**c–f**) were segmented and rendered with VG Studio Max and were shown in a perspective view without a scale bar. AS atrial siphon, CG central ganglion, DT dorsal tubercle, OS oral siphon, OT oral tentacle.

are widely used as indicators of environmental stressors, including noise[49,50], temperature[51], and water quality[52,53].

Prior work has already linked elevated temperature to stress responses in *H. papillosa*[51]. Because ascidians are filter feeders, their physiological condition and settlement behavior are valuable bioindicators[53].

HiTT imaging proved particularly effective for visualizing small, soft-tissue structures without the need for invasive preparation. It has been previously demonstrated that HiTT produces high-resolution images in bivalves[54]. We now demonstrate its applicability to ascidians and its capacity to resolve delicate structures such as oral tentacles and the dorsal tubercle. The resulting 3D renderings provide precise spatial information on the position and connectivity of organs.

Further, we implemented non-destructive imaging techniques, such as MRI and HiTT, to visualize the entire animal as well as selected sections in high resolution.

Finally, our findings highlight the limitations of relying exclusively on model organisms[19]. Even closely related species can differ markedly in anatomy and associated functions. *H. papillosa*, endemic to the Mediterranean Sea and Eastern Atlantic[55], is abundant, ecologically important, and economically relevant[56], yet remains understudied. Our results demonstrate that conclusions drawn from anatomy-based studies of distantly related taxa should be applied cautiously.

## Materials and methods
### Study organism

*Halocynthia papillosa* is a common solitary ascidian found in the Mediterranean Sea. According to the Society for Laboratory Animal Science (GV-SOLAS), no animal testing application was necessary since the animal used is an invertebrate (non-cephalopod). However, the experiments were conducted only by individuals trained and educated in animal experimentation

**Fig. 9 | Tentacle visualization.** Different sections of the oral tentacles. **a** Upper part of a sea squirt with oral (OS) and atrial siphon (AS). **a')** Magnified part of **a** with more details in the sub-tentacle structure of the tentacles. **b** Top view of a 3D-rendered tentacle image from HiTT. **c)** Same tentacle from below. **d, e** 3D image of the blood vessels (red) and the nerval structure (green). While (**d**) only shows vessels and nerves, (**e**) shows half of the surrounding tissue. Black outlined arrowheads in (**d**) show the blood vessels of the second order that supply single sub-tentacles. Black arrowheads show nerves of the second order that innervate the sub-tentacles of the first order from the main nerve. Images (**b–d**) were segmented and rendered with VG Studio Max. While (**b**) and (**c**) represent a parallel view, including a scale bar, (**d**) and (**e**) represent a perspective view, so no scale bar is shown. AS, atrial siphon; OS, oral siphon.

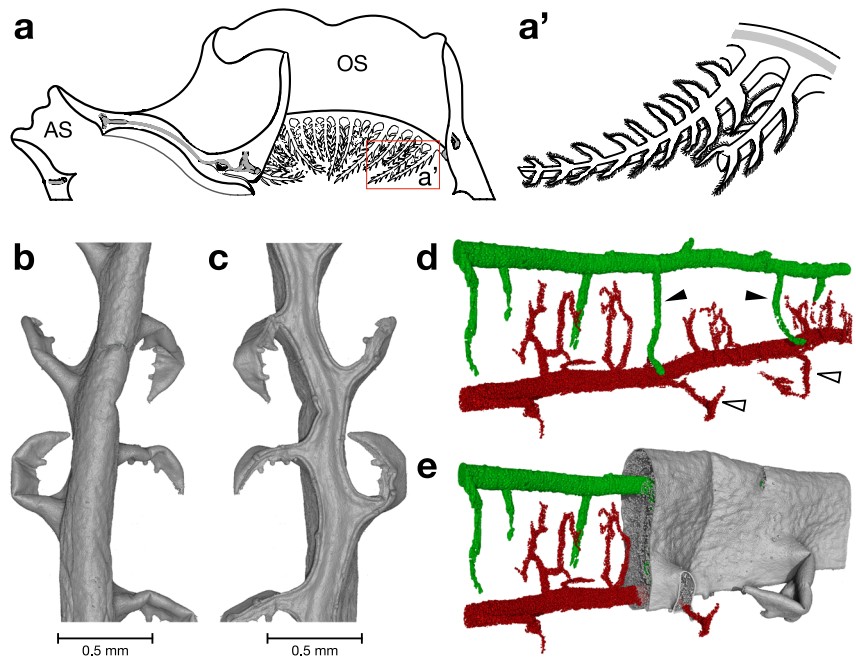

(EU function A). All experiments were performed to the best of our knowledge and in a manner intended to minimize stress and suffering to the animals. All invasive procedures were carried out only after anesthesia. Anesthesia was induced by incubating the animals with menthol crystals for 30 min, which are known to act as mild muscle relaxants in marine invertebrates[57].

### Animal transportation

The organisms were retrieved from the Adriatic Sea near Pula on August 21st, 2022, and on June 15th, 2025 (animals for tunic spectral measurements) at a depth ranging from 10 to 25 meters through scuba diving. After consultations with the Croatian Ministry of Environmental and Nature Protection, it was determined that a sampling permit was not required for ascidians at the designated collection site near Pula (N 44.83 °, E 13.84 °). The collection process was carried out with precision, using a knife to carefully detach the substrate to which the organisms were adhered, thereby minimizing any potential injury to the specimens. Subsequent transportation to the aquatic laboratory at Ruhr University Bochum (RUB) was achieved using plastic containers fitted with individual ventilation via air stones and provisions for continuous cooling to maintain a temperature of 16 °C.

### Animal care and husbandry

Ascidians were placed in an aquarium (123 L volume) at a density of less than one animal per two liters of seawater. Continuous water flow with a turnover rate of 7–9 was achieved by an EHEIM compactON 1000 pump (15 W). This setup was enhanced by an external chilling unit (TECO TK150, maintaining a temperature of 16 °C) to ensure optimal thermal conditions. A comprehensive filtration system included an active carbon filter (Aqua-Medic carbolit 4 mm), a biological filter using plastic balls (AquaMedic miniballs), and a ceramic filter (AquaNova NCR-0.5). Feeding protocols consisted of six daily feeds, providing a total of 20 ml ( ~ 2 × 108 cells) of *Nannochloropsis salina* per ascidian each day. Clean artificial seawater (AquaMedic Reef Salt) with a salinity of 37–39 PSU was supplied through a 30% water exchange every week. The light cycle was set to 12 h/12 h (light/dark) and was provided by an LED lamp (Aquasky, 6500 K, 21 W) integrated into the aquarium.

### Light microscopy

A Leica EZ4 was used for light microscopic examinations with a specimen number of $n = 10$. The selected lens configuration permits magnifications ranging from 8x to 35x, enabling various illumination techniques, including transmitted light, top light, and side light. This flexibility provided a thorough visualization of the diverse tissue specimens. The light microscope primarily captured consecutive images depicting tissues and dissections.

### MRI

Magnetic resonance imaging (MRI) was conducted ex vivo using a 9.4 T horizontal small animal scanner (*BioSpec® 94/20 USR*, Bruker BioSpin GmbH & Co. KG, Germany), equipped with the *B-GA12S HP* gradient system (Bruker BioSpin GmbH & Co. KG, Germany), using the acquisition software ParaVision 360.1.1 at the Leibnitz Institute for Neurobiology in Magdeburg, Germany, on two samples ($n = 2$). A $_1$H transmit-receive volume coil with an inner diameter of 40 mm (Bruker BioSpin GmbH & Co. KG, Germany) was utilized for all measurements. The 4% paraformaldehyde (PFA) pre-fixed *H. papillosa* was incubated in Fomblin® (a hydrogen-free, high-performance precision mechanics pump oil) for two hours to enhance the contrast between the specimen and the surrounding fluid. Based on preliminary experimental testing (not shown here), T2-weighted multi-slice multi-echo (MSME) sequences were found to be the most effective method for visualizing the ascidian.

Before the anatomical scans were conducted, a $B_0$-map was acquired (TR 15 ms, four avg.; image matrix 64', FoV $40 \times 40 \times 40$ mm³). The following imaging parameters were used: Sequence type: MSME sequence; TR: 13921.8 ms; TE: 16.1/80.5/187.7 ms; Echo averages: 2/10/10; transverse imaging (Fig. 2a, b) with 25 contiguous slices of 0.5 mm; sagittal imaging (Fig. 2c, d) with 55 contiguous slices of 0.5 mm; Image size: $3.2 \times 3.8$ cm²; Matrix size: $320 \times 380$ (resolution: $100 \times 100$ µm²); Scan time for transverse images: 0h33m44s; Scan time for sagittal images: 1h28m10s. Isotropic images for 3D visualization (Supplemental Movie 1) were acquired with the same sequence and parameters, but with a matrix size of $320 \times 380$ (resolution: $250 \times 250 \times 250$ µm³) and a scan time of 20 h 41 min. Contrast adjustments of the images were performed using ImageJ, and measurements of dimensions were obtained with Osirix®. 3D visualization and explosion videos were produced with VGStudio Max (2024.4).

## HiTT

High-throughput tomography (HiTT) was conducted at the European Molecular Biology Laboratory (EMBL) beamline P14, utilizing the Petra III storage ring at the Deutsches Elektronen-Synchrotron (DESY) in Hamburg[58]. Given the constraints imposed by a field of view (FOV) limited to a width of 1.3 mm, the dissection of the ascidian was undertaken to isolate specific sections of interest. These included the oral tentacles ($n = 5$), the complete nerve between the two siphons ($n = 1$), a 1 mm punch biopsy of tunic tissue ($n = 5$) near the oral siphon, and muscle strands between the siphons ($n = 3$). The isolated segments were subsequently fixed in 4% PFA overnight, followed by a series of increasing alcohol concentrations according to the protocol established by Zhanmu et al. (2020)[59] to achieve a terminal concentration of 99.8% ethanol, thereby enhancing image contrast in the HiTT data. After preparation, the samples were carefully placed into a 200 µm pipette tip, sealed at the end, fixed to a magnetic goniometer base (MiTeGen Type B5), and covered with the corresponding magnetic lid. The samples were loaded into SPINE pucks and then placed into a sealed sample dewar maintained at room temperature within the beamline hutch. A computer-controlled robotic arm (MARVIN[60]) was used to mount the samples onto the diffractometer. The optimal scanning position was precisely determined using an integrated light microscope within the diffractometer setup (Arinax MD3). The X-ray energy at beamline P14 is adjustable within the range of 7-30 keV, with our measurements conducted at 12.7 keV. A 10-fold magnifying objective was used for each scan, resulting in an effective voxel size of 650 nm. Samples were imaged at four different propagation distances of 73 mm, 77 mm, 83 mm, and 92 mm. For each distance, a total of 1810 projection images were obtained over an 181° rotation angle, along with 100 additional flat-field frames[58]. The total acquisition time for all four distances was 136 s. Following the acquisition phase, reconstruction was completed fully automatically within one minute using the in-house TOMO-CTF software package[61]. Phase retrieval was performed using the contrast-transfer function approach, employing a beta-delta ratio of 0.1. In the case of larger samples, a tiled acquisition of multiple adjacent data sets was performed. Reconstructed volumes were stitched together later using NRStitcher[62]. Data analysis, including segmentation and 3D visualization, was conducted using VG Studio Max (Version 2022.4, 64-bit) following the methodologies outlined in Albers et al. (2024)[58].

## Confocal microscopy

Confocal microscopy was performed at RUB using a Leica TCS SP5 system. For this analysis, the tunic tissue specimens ($n = 5$) were fixed in a 4% paraformaldehyde (PFA) solution and subsequently embedded in Tissue-Tec®. Tissue slicing for visualizing cuticular sheds was carried out with a CM 3050 S microtome, producing sections with a thickness of 50 µm. The imaging process involved positioning the tissue slices, which were affixed to super frost sample holders and covered with a cover slip, in an inverted orientation within the microscope. A 40x objective lens was utilized for imaging CS sections, while a 63x oil immersion lens provided enhanced magnification. To generate three-dimensional maximum intensity projections of the spines with significant depth and minimal background noise, virtual slice stacks comprising up to 101 optical planes were recorded. Additional CS sections were imaged in a non-fixed state to confirm further the absence of autofluorescence induced by PFA fixation. All acquired images relied solely on the tissue's inherent autofluorescence without introducing any artificial fluorescence. The GFP laser used for imaging was a DPSS 561.

## Thunder

Thunder imaging was conducted at RUB using a Leica microscope (model M205 FCA) equipped with a DFC9000 GT camera. This advanced microscopy system offers a magnification range from 0.75 to 16.0x, facilitating extensive imaging applications. Thunder imaging techniques were employed to create whole-animal ($n = 5$) images and perform initial assessments of auto-fluorescence within the CS at low magnification in the living specimen (in vivo). Throughout the study, the straightforward microscope design proved most suitable for two-dimensional imaging with DAPI staining, as light and filters can be easily adjusted. Additionally, conventional mounting fluids were replaced with Roti®-Mount FluorCare DAPI solution (Roth) for tissue sample preparation, allowing for image acquisition with a high signal-to-noise ratio.

## FTIR spectroscopy

Fourier-transform infrared spectroscopy was performed at the Chair of Applied Electrodynamics and Plasma Technology (AEPT) at RUB, using a Bruker Hyperion 3000 equipped with an Infinity 1 video camera. The tunic ($n = 2$) was sliced immediately after anesthesia using a vibratome (Leica VT 1200) without any fixing detergents in artificial salt water (38 PSU). The resulting slice thickness was calibrated to 10 µm. The specimens were promptly transferred to silicon wafers for drying following the slicing procedure. Visual inspection under a light microscope confirmed that no salt crystals formed on the surface during drying. The samples underwent overnight incubation before measurement. In a nitrogen-precooled scanning environment, the scanning process utilized an LN-MCT-D316-025 detector with a total scanning duration of 14 min and 17 s. It included 32 sample scans and 32 background scans in the microscopic scanning position.

## Spectral measurements

The spectral measurements were carried out at RUB on 12 adult individuals of *H. papillosa*. For each animal, four measurements were taken on defined body regions: on the red and on the light-colored side of the body, each in the contracted and in the relaxed state. The measurements were carried out under controlled conditions in a darkened room. The optical sensor was constantly positioned at a right angle to the sample surface and kept at a distance of 4 mm to ensure reproducible conditions.

To determine the center of the body on both sides, an imaginary line was drawn from the oral siphon from anterior to posterior (coronal plane), and an additional line was drawn from the atrial siphon from dorsal to ventral. The point where the lines crossed was defined as the center of the body.

An Ocean Optics spectrometer (model FLMS05961) was used in combination with the OceanView Spectroscopy software (version 2.0.19-rcl) for data acquisition. The spectrometer settings included a trigger mode on software triggering (On Demand), an integration time of 1000 microseconds, averaging over 20 scans, an activated electrical dark correction, an activated non-linearity correction, and a boxcar smoothing with a width of three. The X-axis was displayed in wavelength mode, and the recorded spectrum consisted of 2048 pixels. The measurement software ran under Mac OS X version 15.6 with Java Runtime Environment version 1.8.0_461.

## Statistics and reproducibility

The recorded raw data of the spectral measurements was analyzed in R (version 2024.12.0 + 467)[63]. After importing and formatting, the spectral intensity curves were first averaged for each condition and body region combination. The spectral peaks were identified from these averaged curves. The individual intensity values per animal were extracted at the corresponding wavelengths. Spectral maxima were checked for normal distribution with Shapiro test and QQ plots in R and were square root transformed to meet assumptions of normality. Homogeneity of variances was tested with a Levene's test. A two-factorial ANOVA with repeated measures (animal ID) was conducted to analyze the main effects of condition and body region, as well as their interaction. This was followed by a one-way ANOVA with repeated measures (animal ID) to assess the effect of condition by keeping body region constant. Bonferroni correction was applied. All statistical analyses were performed using the packages car[64], rstatix[65], dplyr[66], tidyr[67], tidyverse[68], ggplot2[69], lsr[70], plyr[71] and stats[63]. For the imaging techniques that did not produce any numerical data and, therefore, statistical testing was not conducted, sample units and sizes were as follows:

- MRI: sample unit = whole individual ascidian, sample size = 1

<div style="text-align: right"><strong>Article</strong></div>

- Tunic structure overview (Light microscopy): sample unit = tunic slices from different individual ascidians, sample size = 5
- Tunic fluorescence (Thunder microscopy): sample unit = tunic of different individual ascidians, sample size = 5
- Tunic structure and fluorescence (Confocal microscopy): sample unit = tunic slices from different individual ascidians, sample size = 5
- Tunic structure (HiTT): sample unit = tunic sections from different individual ascidians, sample size = 5
- Tunic nuclei distribution (DAPI staining): sample unit = tunic slices from different individual ascidians, sample size = 5
- Tunic composition (FTIR): sample unit = measurement points within one 10 μm tunic slice from one individual, sample size: 10
- Nervous system (Light microscopy): sample unit = individual ascidian, sample size = 3
- Nervous system (HiTT): sample unit = whole nerve of individual ascidian, sample size = 2
- Nervous system (HiTT): sample unit = nerve sections of individual ascidian, sample size = 5
- Oral tentacle investigation (HiTT): sample unit = oral tentacles of different individual ascidians, sample size = 5

## Reporting summary

Further information on research design is available in the Nature Portfolio Reporting Summary linked to this article.

## Data availability

The numerical source data for the graphs can be found in Supplementary Data 1 and 2.

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

## Acknowledgements

HiTT data were collected on EMBL Beamline P14 on the Petra III synchrotron in Hamburg. We thank Diving Pula for support with logistics. This work was supported by Deutsche Forschungsgemeinschaft (DFG) grants: Project ID 316803389 — SFB 1280, SH (Subproject A07); Project number 492434978 — GRK 2862/1, Sub-projects (01, SH; 09, IS).

## Author contributions

L.H.: Hypothesis generation, conceptualization, literature search, methodology, data scanning, image analysis, data evaluation and analysis, drawing, manuscript writing. J.A.: Methodology, data scanning, image analysis, data evaluation, and analysis. A.M.: Methodology, data scanning, image analysis, data evaluation, and analysis. T.B.: Methodology, data scanning, image analysis, data evaluation, and analysis, manuscript writing. E.D.: Methodology, data scanning. I.S.: Methodology S.H.: Conceptualization, funding acquisition. J.G.: Hypothesis generation, conceptualization, methodology, data scanning, image analysis, manuscript writing. M.H.: Hypothesis generation, conceptualization, methodology, image analysis, data evaluation and analysis, manuscript writing. All authors reviewed the manuscript.

## Funding

## Competing interests

The authors declare no competing interests.
