## [Transparent Peer Review file · Communications Biology]

Insights into unique anatomical structures of the ascidian *Halocynthia papillosa* obtained by multimodal imaging

Corresponding Author: Dr Mareike Huhn

Version 0:

Reviewer comments:

Reviewer #1

(Remarks to the Author)

The manuscript “New insights into unique anatomical structures of the ascidian *Halocynthia papillosa* obtained by multimodal imaging” is well written, clearly structured, and presents its findings in a coherent and accessible manner. The figures are informative, and the methodology is described in sufficient detail to allow reproducibility.

This manuscript presents a detailed anatomical study of *Halocynthia papillosa* using advanced imaging techniques, offering valuable new insights into the structure of the tunic and neural complex. The imaging methods are technically impressive and generate novel, high-resolution datasets that will be useful for future studies. However, the work remains largely descriptive, with no functional experiments to support broader biological conclusions. As such, in its current form, it may fall short of the criteria for Communications Biology, which prioritizes studies that provide significant new biological insight.

If we review the publication criteria outlined by Communications Biology:

- The results are novel – Yes
- The paper provides strong evidence for its conclusions – Yes
- The data are technically sound – Yes
- The manuscript is important to scientists in the specific sub-field of biology – Yes
- The paper should represent an advance in understanding which may influence thinking in the field – Partially

While the paper represents a technical advance, it does not currently provide a conceptual advance. Without either a hypothesis-driven framework, functional experiments, or an extended discussion on how these anatomical datasets could be used to generate new biological insights, it may not fully meet the journal’s expectation of providing an advance in understanding.

I encourage the authors to consider including a more explicit interpretation of how their results advance the field, and if possible, support this with functional data to broaden the biological relevance of their work.

General comments-

The methods section would benefit from specifying the number of specimens used for each technique to clarify the scope and reproducibility of the analyses.

The discussion section is quite lengthy and technical, which makes it difficult to discern the main message. Clarifying the key takeaways and emphasizing the potential applications of the parameters you identified would strengthen the impact of the manuscript.

Specific comments-

Line 44- “highly developed immune systems. As far as I know the immune system of ascidians is not highly developed”.

Please add references that support this claim.

Line 84- “The present study aimed to investigate various sections of the *H. papillosa* tunic and the neural structures thought to be linked to sound perception and general neural processing”.

It would be helpful if the authors could clarify whether and how their anatomical findings can contribute to new insights into neural processing. As currently presented, the study provides detailed anatomical descriptions, but the link to functional neural mechanisms remains speculative. Expanding on this point could strengthen the impact of the work. Same for line 379 in the discussion.

Line 129- “The integration of various imaging techniques (x-ray, light 130 microscopy, DAPI staining) enabled the creation of a comprehensive scheme summarizing all the information (Fig. 3 A). The appearance of the CS and MZ varied among samples from different individuals and across different sections of the same individual. In tunic sections that had not been exposed to sunlight (left or right side of the animal, depending on its orientation in the reef, and animals collected from caves or beneath overhangs), the MZ appeared pale white with barely distinguishable tunicin layers”.

The mention of tunic differences in animals from shaded environments (e.g., caves, overhangs) suggests the potential for a more comparative approach. How many animals were sampled from these distinct environmental conditions? Could the observed differences in tunicin layer visibility relate to UV exposure or protective functions of the tunic? Additionally, were any corresponding differences observed in the neural structures of animals from different habitats?

Similar to the previous comment, I would suggest clarifying the functional implications of the observations described in line 136 (color variation), line 327 (autofluorescent regions), and line 342 (different developmental stages). Including hypotheses or suggestions for how these findings might be used in future functional studies would enhance the impact of the manuscript and help move it beyond its currently descriptive scope.

Line 260- "All four species investigated that belong to the same order as *H. papillosa* (Stolidobranchia) showed distinct thickening of the nerve at the CG. For the genus *Halocynthia*, only one study (on *H. roretzi*) has published images of the CG, and neither has found any thickening in the central nerve. In the respective study, the authors refer to the entire nerve between the two ends of dichotomous branching as the central ganglion of *H. roretzi*".

Could the authors comment on whether there is published work, or perhaps their own behavioral observations, that relate the absence of a defined CG to neural processing or activity patterns in *H. papillosa*?

Line 268- "We, therefore, suggest that the cerebral ganglion only comprises a subsection of the nerve between the dichotomous branches and is visually not distinct".

This sentence is a bit unclear. Are you suggesting that *H. papillosa* does possess a cerebral ganglion, but that it is not anatomically distinguishable from the surrounding nerve tissue? Or are you proposing that this species lacks a cerebral ganglion altogether? Please add images of the relevant area that is proposed to be the CG.

Line 270- "Further investigations, including staining methods such as Nissl staining, MAP2, BP102, and Pan-Nav, should clarify this assumption and are planned for future studies". The planned staining approaches sound very promising and would greatly help clarify the presence and structure of the cerebral ganglion.

Line 284- "...and later detailed in the model organisms *Corella inflata*, *Styela plicata*, and *Ciona robusta*". References are missing.

Line 291- "...coronal organ..."- It is unclear whether the coronal organ was definitively identified in this species based on the current results. If available, including representative images or clarifying its detection would strengthen this section.

Reviewer #2

(Remarks to the Author)

This study used multimodal imaging techniques to analyze the structure of tunicate tissues (tunic, cerebral ganglion, and tentacles). The high-resolution 3D images obtained using the proposed method provide information that cannot be obtained using conventional methods. However, there are insufficient comparisons with morphological findings already described for the same or closely related species. This method is attractive because it allows for the observation of detailed internal structures of specimens in a non- or minimally destructive manner. In particular, the ability to analyze the internal structures of rare specimens in a minimally destructive manner could resolve various issues. Comprehensive structural comparisons across a wide range of taxonomic groups are expected to yield new insights. This point can be further emphasized.

A correct overview of the basic structure of the tunic is necessary in the Introduction. The tunic is an extracellular matrix composed mainly of cellulose, which is synthesized by the epidermis. It is located outside the epidermis and various types of cells are distributed within the matrix to perform different functions. These features are unique to the metazoans. Tunicate cellulose nanocrystals have been the subject of numerous studies in the field of material engineering. The following references may be useful.

Burighel P and Cloney RA (1997) Urochordata: Ascidiacea. In Harrison F W and Ruppert E (eds.) Microscopic anatomy of invertebrates, Vol. 15. Hemichordata, Chaetognatha, and the invertebrate chordates, pp. 221–347. New York: Wiley-Liss, Inc.

Hirose (2009) Invertebrate Biology 128: 83–96.

Kimura & Itoh (2004) Cellulose 11: 377–383.

Matthysse et al. (2004) Proc Natl Acad Sci USA 101: 986–991

Nakashima et al. (2004) Dev Genes Evol 214: 81–88

In the text, the fibers of the tunic are referred to as tunicins, but this term is rarely used today. Tunicin was a term used to distinguish cellulose-like polysaccharide fibers in ascidians from plant cellulose in the 70s, but it has been proven that the fibers of the capsule are mainly composed of pure cellulose.

Similarly, "zooid" is a term used to refer to individuals within a colony in colonial species, and is not appropriate for use with solitary species.

To demonstrate the validity of the present observation methods, it is necessary to compare what has become apparent through this method, whether there are any structures that cannot be observed in the structures described in previous studies, and whether the structures observed in both cases are consistent. Thus, the findings of this study must be compared with those of histological and ultrastructural studies of *Halocynthia* spp.

For example,

Figures 4b and 4c are nearly identical to the scanning electron microscopy (SEM) images in Figure 8d of Song et al. (2020), although this similarity is not noted. Various cell types are scattered within the tunic (e.g., Van Daele et al., 1991, Journal of Structural Biology, 106, 115–12; Song et al., 2020; Hirose et al., 2009). Journal of Fish Diseases 32, 433–445; however, comparison with DAPI observations is insufficient. The structure referred to as "cuticular shed (CS)" is described as being

scattered on the tunic surface in this report. This can be interpreted as the thickened part of the cuticle layer. The cuticle serves as a physical barrier that prevents microbial invasion into the tunic and covers the entire surface of the tunic matrix. In addition, the cuticle layer can regenerate even after damage (Nakayama et al., 2024. *Diseases of Aquatic Organisms*, 159:37–48).

Figures are cited as A, B, C, in the main text, but labels and annotation in the figure captions are a, b, c,..... Which are correct?

The author states, “The present study highlights two structural adaptations that may enhance the ability to perceive sound or vibrations” (lines 378–379), but the specific location is unclear.

The latter part of the Discussion (lines 393–416) is redundant and not directly related to the findings of this study. Thus, this should be reduced.

As mentioned above, many points in the text need to be revised, particularly regarding the appropriate use of terminology. We believe that this article needs to be revised before it can be published in *Communications Biology*. Comments on these points have been added to the manuscript PDF. Please refer to the attached file.

Reviewer #3

(Remarks to the Author)

I co-reviewed this manuscript with one of the reviewers who provided the listed reports. This is part of the *Communications Biology* initiative to facilitate training in peer review and to provide appropriate recognition for Early Career Researchers who co-review manuscripts.

Version 1:

Reviewer comments:

Reviewer #2

(Remarks to the Author)

Although the author has addressed most of the reviewers' comments, several points still require revision. In the previous comments from me, I wrote “Comments on these points have been added to the manuscript PDF. Please refer to the attached file.

However, the author did not address any of the comments noted in the attached file. Did the author not receive the file from the editor? The author is not required to respond to every comment, but please be sure to review them as they include incorrect citations and instructions for adding scales.

I will attach the same file as the last time.

Line 37–38: “Besides the tunic and the two siphons, which serve as a physical barrier”. Two siphons would not work as a physical barrier.

Line 63: “ascidiacea, thaliacea, and appendicularia” should be “Ascidiacea, Thaliacea, and Appendicularia”. They should not be italicized.

Line 151: “Fig. 9”. Figure numbers should be consistent with the order in which they appear in the manuscript.

Line 154: “ $F(1,12) = 75.99, p < 0.001$ ” and Line 158: “dark body side: $t(12) = 7.24, p < 0.00$ ”. The meaning of F and t should be explained.

Line 166: “(Fig. f&i)” should be “(Fig. 3 f&i)”.

Line 223: “Fig. 6 e-f”. The hyphen should be changed to en dash like Line 240. This mistype emerged in all the manuscript.

Line 240: “(7 h-i)” should be (Fig. 7 h-i).

Line 284: “Repeated dichotomous branching - which has not been reported for Stolidobranchia19- was observed in *H. papillosa*”. In literature, tentacle shape of this species is described briefly as below. You can cite this. “The branchial tentacles, which are not very bushy, have long primary branches, short secondary branches and minute tertiary branches” Kott, P. (1985). *The Australian Ascidiacea. Part 1, Phlebobranchia and Stolidobranchia*. *Memoirs of the Queensland Museum* 23: 1-440., available online at <http://biostor.org/reference/109626> (p. 344)

Line 290–291: “*H. papillosa* is the only known ascidian with branched oral tentacles.: is not correct. Branching oral (or branchial) tentacle is common character states in Pyuridae and Molguridae. (see Kott, 1985 mentioned above).

Line 307, 309: “spp.” should be roman

Line 310–311: It is inaccurate to definitively state that “which bear spines absent in *H. roretzi*” The presence of spines likely

varies depending on the observation site. Spines are observed on the sac surrounding the siphon in *H. roretzi*. See Fig. 4a in Kitamura et al. 2010. Tunic morphology and viral surveillance in diseased Korean ascidians: Soft tunic syndrome in the edible ascidian, *Halocynthia roretzi* (Drasche), in aquaculture. *Journal of Fish Diseases* 33: 153–160. Fig. 4a

Line 364: “only after anesthesia”. Which type of anesthesia was used? It would be better to describe it in more detail.

References:

Species name (latino names) are roman in the list but they should be italic. (e.g. *Corella inflata* ; *Ciona intestinalis*,)
Some journal titles do not capitalize the first letter of each word. (ex. Canadian journal of zoology; Zoological science.....)
Incorrect DOI URL: repeated “https” like “https://doi.org/https://doi.org”.

Figures:

Figure 1 should be cited in the manuscript.

In Figure 1, it is difficult to distinguish the "Small brown arrows" and the "green arrows".

In Figure 3b–c, scale bar or magnification should be indicated.

Reviewer #3

(Remarks to the Author)

I co-reviewed this manuscript with one of the reviewers who provided the listed reports. This is part of the Communications Biology initiative to facilitate training in peer review and to provide appropriate recognition for Early Career Researchers who co-review manuscripts.

Response to reviewers

Reviewer #1 (Remarks to the Author):

The manuscript “New insights into unique anatomical structures of the ascidian Halocynthia papillosa obtained by multimodal imaging” is well written, clearly structured, and presents its findings in a coherent and accessible manner. The figures are informative, and the methodology is described in sufficient detail to allow reproducibility. This manuscript presents a detailed anatomical study of Halocynthia papillosa using advanced imaging techniques, offering valuable new insights into the structure of the tunic and neural complex. The imaging methods are technically impressive and generate novel, high-resolution datasets that will be useful for future studies. However, the work remains largely descriptive, with no functional experiments to support broader biological conclusions. As such, in its current form, it may fall short of the criteria for Communications Biology, which prioritizes studies that provide significant new biological insight. If we review the publication criteria outlined by Communications Biology:

- *The results are novel – Yes*
- *The paper provides strong evidence for its conclusions – Yes*
- *The data are technically sound – Yes*
- *The manuscript is important to scientists in the specific sub-field of biology – Yes*
- *The paper should represent an advance in understanding which may influence thinking in the field – Partially*

While the paper represents a technical advance, it does not currently provide a conceptual advance. Without either a hypothesis-driven framework, functional experiments, or an extended discussion on how these anatomical datasets could be used to generate new biological insights, it may not fully meet the journal’s expectation of providing an advance in understanding. I encourage the authors to consider including a more explicit interpretation of how their results advance the field, and if possible, support this with functional data to broaden the biological relevance of their work.

General comments-

1.1 Reviewer comment:

The methods section would benefit from specifying the number of specimens used for each technique to clarify the scope and reproducibility of the analyses.

1.1 Response:

Thank you for the notice. We added the number of specimens in the methods section of the manuscript separately for each used technique.

1.2 Reviewer comment:

The discussion section is quite lengthy and technical, which makes it difficult to discern the main message. Clarifying the key takeaways and emphasizing the potential applications of the parameters you identified would strengthen the impact of the manuscript.

1.2 Response:

We have revised the entire discussion and fused it with the conclusion. We now clearly emphasize the key takeaways and have added more suggestions for future studies and applications that could be conclusive based on our findings. We removed redundant sections and restructured the discussion, especially in the section about the tunic (lines 266-353).

Specific comments-

1.3 Reviewer comment:

Line 44- “highly developed immune systems. As far as I know the immune system of ascidians is not highly developed”. Please add references that support this claim.

1.3 Response:

Thank you for this remark. We rephrased the sentence and replaced the term with one more accurately reflecting our intended meaning (lines 37-41):

1.4 Reviewer comment:

Line 84- “The present study aimed to investigate various sections of the H. papillosa tunic and the neural structures thought to be linked to sound perception and general neural processing”. It would be helpful if the authors could clarify whether and how their anatomical findings can contribute to new insights into neural processing. As currently presented, the study provides detailed anatomical descriptions, but the link to functional neural mechanisms remains speculative. Expanding on this point could strengthen the impact of the work. Same for line 379 in the discussion.

1.4 Response:

Thank you for the suggestion. In the revised manuscript, we have removed the emphasis on sound perception in the introduction (lines 97-100). Even though one of the aims of the study was to have a closer look at anatomical structures involved in sound processing, their exact function remains speculative. We, therefore, decided to focus on the detailed description of the structures and suggestions on how future studies could use the results presented in this study for obtaining a better understanding of their involvement in sound perception and neural processing (lines 335-341). This also helped in shortening and focusing the discussion as suggested.

1.5 Reviewer comment:

Line 129- “The integration of various imaging techniques (x-ray, light 130 microscopy, DAPI staining) enabled the creation of a comprehensive scheme summarizing all the information (Fig. 3 A). The appearance of the CS and MZ varied among samples from different individuals and across different sections of the same individual. In tunic sections that had not been exposed to sunlight (left or right side of the animal, depending on its orientation in the reef, and animals collected from caves or beneath overhangs), the MZ appeared pale white with barely distinguishable tunicin layers”. The mention of tunic differences in animals from shaded environments (e.g., caves, overhangs) suggests the potential for a more comparative approach. How many animals were sampled from these distinct environmental conditions? Could the observed differences in tunicin layer visibility relate to UV exposure or protective functions of the tunic? Additionally, were any corresponding differences observed in the neural structures of animals from different habitats? Similar to the previous comment, I would suggest clarifying the functional implications of the observations described in line 136 (color variation), line 327 (autofluorescent regions), and line 342 (different developmental stages). Including hypotheses or suggestions for how these findings might be used in future functional studies would enhance the impact of the manuscript and help move it beyond its currently descriptive scope.

1.5 Response:

Thank you very much for pointing out that important point. While writing this manuscript, we decided to keep the information about the color differences of the animals' tunics in different habitats for a comprehensive future study. However, as you pointed out, it would help to include a short section with clarifying results about the color differences of the tunic. Based on the expenditure necessary to perform a comprehensive study about the different habitats and color adaptations of *H. papillosa*, we decided to focus on the color variations within the animal, for now, and have added data on spectral intensities of red colors on the light (light averted) and dark (light exposed) side of the animal. Using an underwater spectrometer, we measured the intensity spectra of the light and dark sides in 13 animals. This examination led to noticeable results that are now presented and discussed in the respective sections of methods (lines 493-502), results (lines 146-159, figure 9), and discussion (lines 320-325).

We also modified the paragraph in which we discuss the fluorescence in the tunic and now cite literature that has investigated functions of fluorescence in marine organisms and a previous study that found fluorescence in an ascidian (lines 316-319).

1.6 Reviewer comment

*Line 260- “All four species investigated that belong to the same order as *H. papillosa* (Stolidobranchia) showed distinct thickening of the nerve at the CG. For the genus *Halocynthia*, only one study (on *H. roretzi*) has published images of the CG, and neither has found any thickening in the central nerve. In the respective study, the authors refer to the entire nerve between the two ends of dichotomous branching as the central ganglion of *H. roretzi*”. Could the authors comment on whether there is published work, or perhaps their own behavioral observations, that relate the absence of a defined CG to neural processing or activity patterns in *H. papillosa*? Line 268- “We, therefore, suggest that the cerebral ganglion only*

comprises a subsection of the nerve between the dichotomous branches and is visually not distinct". This sentence is a bit unclear. Are you suggesting that H. papillosa does possess a cerebral ganglion, but that it is not anatomically distinguishable from the surrounding nerve tissue? Or are you proposing that this species lacks a cerebral ganglion altogether? Please add images of the relevant area that is proposed to be the CG.

1.6 Response:

Thank you for pointing out that this section is unclear. Since both comments are regarding the thickening/abundance of the CG, we will answer both in one comment in the following.

We adjusted the sentence (now lines 278-279) to clarify that we suggest that the investigated species *H. papillosa* has a CG, but that it is not definitely distinguishable with the methods used. Nevertheless, we point out that the species is the first to have such a long central nerve between the two dichotomous branches (lines 276-278).

1.7 Reviewer comment

Line 270- "Further investigations, including staining methods such as Nissl staining, MAP2, BP102, and Pan-Nav, should clarify this assumption and are planned for future studies". The planned staining approaches sound very promising and would greatly help clarify the presence and structure of the cerebral ganglion.

1.7 Response:

We will perform a comprehensive study examining the abundance and position of the CG in *Halocynthia papillosa* in a future study, but it would have exceeded the scope of this study.

1.8 Reviewer comment:

Line 284- "...and later detailed in the model organisms Corella inflata, Styela plicata, and Ciona robusta". References are missing.

1.8 Response:

Thanks for this remark. We added the references in the manuscript (lines 287-288).

1.9 Reviewer comment:

Line 291- "...coronal organ..."- It is unclear whether the coronal organ was definitively identified in this species based on the current results. If available, including representative images or clarifying its detection would strengthen this section.

1.9 Response:

We were able to identify the position of the coronal organ using the HiTT imaging. The resolution (approx. 650nm) was not sufficient to identify the individual hair cells (~300nm) within the coronal organ, but blurry outlines of hair cells were visible. These images were not clear enough to add as images to the manuscript.

Reviewer #2 (Remarks to the Author):

This study used multimodal imaging techniques to analyze the structure of tunicate tissues (tunic, cerebral ganglion, and tentacles). The high-resolution 3D images obtained using the proposed method provide information that cannot be obtained using conventional methods. However, there are insufficient comparisons with morphological findings already described for the same or closely related species. This method is attractive because it allows for the observation of detailed internal structures of specimens in a non- or minimally destructive manner. In particular, the ability to analyze the internal structures of rare specimens in a minimally destructive manner could resolve various issues. Comprehensive structural comparisons across a wide range of taxonomic groups are expected to yield new insights. This point can be further emphasized.

2.1 Reviewer comment:

A correct overview of the basic structure of the tunic is necessary in the Introduction. The tunic is an extracellular matrix composed mainly of cellulose, which is synthesized by the epidermis. It is located outside the epidermis and various types of cells are distributed within the matrix to perform different functions. These features are unique to the metazoans. Tunicate cellulose nanocrystals have been the subject of numerous studies in the field of material engineering. The following references may be useful. Burighel P and Cloney RA (1997) Urochordata: Ascidiacea. In Harrison F W and Ruppert E (eds.) Microscopic anatomy of invertebrates, Vol. 15. Hemichordata, Chaetognatha, and the invertebrate chordates, pp. 221–347. New York: Wiley-Liss, Inc. Hirose (2009) Invertebrate Biology 128: 83–96. Kimura & Itoh (2004) Cellulose 11: 377–383. Matthyse et al. (2004) Proc Natl Acad Sci USA 101: 986–991 Nakashima et al. (2004) Dev Genes Evol 214: 81–88

2.1 Response:

Thank you very much for pointing out that a general introduction to the tunic is necessary in the beginning. We added an informative introduction regarding the anatomy and synthesis of the tunic in ascidians, including validating references. Further, we clarified the terms tunicin and cellulose and their use in the manuscript (lines 60-68).

2.2 Reviewer comment:

In the text, the fibers of the tunic are referred to as tunicins, but this term is rarely used today. Tunicin was a term used to distinguish cellulose-like polysaccharide fibers in ascidians from plant cellulose in the 70s, but it has been proven that the fibers of the capsule are mainly composed of pure cellulose.

2.2 Response:

Thank you for this remark. In our understanding, the term cellulose still differs slightly from the term tunicin. In fact, tunicin mainly consists of cellulose. However, the structure shows differences, as tunicin is composed of crystalline cellulose fibers and contains specific proteins that make the structure unique

and therefore different from cellulose. Also, it is still used by some authors of papers published in the last 10 years (<https://www.frontiersin.org/journals/immunology/articles/10.3389/fimmu.2017.00674/full>), although the term cellulose is mainly used. To understand the term tunicin compared to cellulose, we decided to clarify the terms cellulose and tunicin at the beginning of the introduction and use the term cellulose throughout the manuscript (lines 67-68).

2.3 Reviewer comment:

Similarly, "zooid" is a term used to refer to individuals within a colony in colonial species, and is not appropriate for use with solitary species.

2.3 Response:

Thank you for pointing this out. We changed the term accordingly throughout the manuscript.

2.4 Reviewer comment:

*To demonstrate the validity of the present observation methods, it is necessary to compare what has become apparent through this method, whether there are any structures that cannot be observed in the structures described in previous studies, and whether the structures observed in both cases are consistent. Thus, the findings of this study must be compared with those of histological and ultrastructural studies of *Halocynthia* spp. For example, Figures 4b and 4c are nearly identical to the scanning electron microscopy (SEM) images in Figure 8d of Song et al. (2020), although this similarity is not noted. Various cell types are scattered within the tunic (e.g., Van Daele et al., 1991, *Journal of Structural Biology*, 106, 115–12; Song et al., 2020; Hirose et al., 2009). *Journal of Fish Diseases* 32, 433–445); however, comparison with DAPI observations is insufficient. The structure referred to as "cuticular shed (CS)" is described as being scattered on the tunic surface in this report. This can be interpreted as the thickened part of the cuticle layer. The cuticle serves as a physical barrier that prevents microbial invasion into the tunic and covers the entire surface of the tunic matrix. In addition, the cuticle layer can regenerate even after damage (Nakayama et al., 2024. *Diseases of Aquatic Organisms*, 159:37–48).*

2.4 Response:

Thank you for pointing out that the comparison was not sufficiently explained in the manuscripts so far. We added sections, including relevant references, to further compare our findings with the present literature.

2.5 Reviewer comment:

Figures are cited as A, B, C, in the main text, but labels and annotation in the figure captions are a, b, c,..... Which are correct?

2.5 Response:

Thank you for pointing out that inconsistency. We changed the names in the text throughout the manuscript.

2.6 Reviewer comment:

The author states, “The present study highlights two structural adaptations that may enhance the ability to perceive sound or vibrations” (lines 378–379), but the specific location is unclear.

2.6 Response:

Thank you for this remark. We changed the section in the manuscript, described the two mentioned structures in greater detail, but also decided to move the focus away from the perception of sound and vibrations as suggested by the other reviewer (see response 1.4).

2.7 Reviewer comment:

The latter part of the Discussion (lines 393–416) is redundant and not directly related to the findings of this study. Thus, this should be reduced.

2.7 Response:

We have revised the discussion as already pointed out in our response to reviewer 1. We believe that the revised discussion is not redundant anymore and better emphasizes the takeaway messages of our findings (lines 267-352).

2.8 Reviewer comment:

As mentioned above, many points in the text need to be revised, particularly regarding the appropriate use of terminology. We believe that this article needs to be revised before it can be published in Communications Biology. Comments on these points have been added to the manuscript PDF. Please refer to the attached file.

2.8 Response:

Thank you for the feedback. We have revised with respect to all points and comments made.

Response to reviewers, Second review round, Reviewer comment numbering

Reviewer #2 (Remarks to the Author)

Although the author has addressed most of the reviewers' comments, several points still require revision. In the previous comments from me, I wrote "Comments on these points have been added to the manuscript PDF. Please refer to the attached file. However, the author did not address any of the comments noted in the attached file. Did the author not receive the file from the editor? The author is not required to respond to every comment, but please be sure to review them as they include incorrect citations and instructions for adding scales. I will attach the same file as the last time.

1) Reviewer Comment

Line 37–38: "Besides the tunic and the two siphons, which serve as a physical barrier". Two siphons would not work as a physical barrier.

1) Response:

By closing the siphons, a physical barrier can be created, especially against parasites and predators that attempt to enter the animal's body. To support this statement, we have added an additional reference to the manuscript and have rephrased the sentence for more clarity.

Hoyle, G. (1952). The response mechanism in ascidians. *Journal of the Marine Biological Association of the United Kingdom*, 31(2), 287-305.

2) Reviewer comment:

Line 63: "ascidiacea, thaliacea, and appendicularia" should be "Ascidiacea, Thaliacea, and Appendicularia". They should not be italicized.

2) Response:

Thank you for your feedback. We have made the necessary changes to the manuscript.

3) Reviewer comment:

Line 151: "Fig. 9". Figure numbers should be consistent with the order in which they appear in the manuscript.

3) Response:

Thank you very much for this remark. We forgot to rearrange the Figures after adding Figure 9. We have now changed the order of the Figure numbers so they are named according to their order in the manuscript.

4) *Reviewer comment:*

Line 154: “ $F(1,12) = 75.99, p < 0.001$ ” and Line 158: “dark body side: $t(12) = 7.24, p < 0.00$ ”. The meaning of F and t should be explained.

4) Response:

Thank you for the comment. We have now explained the two values briefly in the manuscript (now lines 160-161 and 165-166).

5) *Reviewer comment:*

Line 166: “(Fig. f&i)” should be “(Fig. 3 f&i)”.

5) Response:

We have added the Figure number in the manuscript.

6) *Reviewer comment:*

Line 223: “Fig. 6 e-f”. The hyphen should be changed to en dash like

Line 240. This mistype emerged in all the manuscript.

6) Response:

Thanks for the remark. All figure links have been changed.

7) *Reviewer comment:*

Line 240: “(7 h-i)” should be (Fig. 7 h-i).

7) Response:

We added “Fig.” in the manuscript

8) *Reviewer comment:*

Line 284: "Repeated dichotomous branching - which has not been reported for *Stolidobranchia*19- was observed in *H. papillosa*". In literature, tentacle shape of this species is described briefly as below. You can cite this. "The branchial tentacles, which are not very bushy, have long primary branches, short secondary branches and minute tertiary branches" Kott, P. (1985). *The Australian Ascidiacea. Part 1, Phlebobranchia and Stolidobranchia. Memoirs of the Queensland Museum 23: 1-440.*, available online at <http://biostor.org/reference/109626> (p. 344)

8) Response:

Thanks for pointing out that dichotomous branching had been observed before. We added the reference in the text section (now line 334).

9) Reviewer comment:

Line 290–291: "*H. papillosa* is the only known ascidian with branched oral tentacles.: is not correct . Branching oral (or branchial) tentacle is common character states in *Pyuridae* and *Molguridae*. (see Kott, 1985 mentioned above).

9) Response

We refined the text section regarding the abundance of dichotomous branching in *H. papillosa* (now lines 341-346).

10) Reviewer comment:

Line 307, 309: "spp." should be roman

10) Response

The changes were made in the manuscript.

11) Reviewer comment:

Line 310–311: It is inaccurate to definitively state that "which bear spines absent in *H. roretzi*" The presence of spines likely varies depending on the observation site. Spines are observed on the sac surrounding the siphon in *H. roretzi*. See Flg. 4a in Kitamura et al. 2010. Tunic morphology and viral surveillance in diseased Korean ascidians: Soft tunic syndrome in the edible ascidian, *Halocynthia roretzi* (Drasche), in aquaculture. *Journal of Fish Diseases* 33: 153–160. Fig. 4a

11) Response:

Thanks for this important remark. We changed the sentence in the manuscript and added the reference (now lines 289-291).

12) Reviewer comment:

Line 364: "only after anesthesia". Which type of anesthesia was used? It would be better to describe it in more detail.

12) Response:

We added a sentence to describe the anesthesia. Also, a reference has been added (now lines 386-388).

13) Reviewer comment:

References:

Species name (latino names) are roman in the list but they should be italic. (e.g. *Corella inflata* ; *Ciona intestinalis*,) Some journal titles do not capitalize the first letter of each word. (ex. *Canadian journal of zoology*; *Zoological science*....) Incorrect DOI URL: repeated "https" like "https://doi.org/https://doi.org/".

13) Response:

Thanks for pointing this out. Changes have been made accordingly.

14) Reviewer comment

Figures:

Figure 1 should be cited in the manuscript.

In Figure 1, it is difficult to distinguish the "Small brown arrows" and the "green arrows".

In Figure 3b–c, scale bar or magnification should be indicated.

14) Response

Figure 1 is now cited in the introduction to visualize the study organism and the general morphological composition. Further, the figure was referenced in the results section as a result of the isotropic MRI. The arrow will be kept in place as the color scheme is used for several other images. Arrows can be clearly distinguished in the high-resolution figure at higher magnification.

Scale bars were added to Figure 3b-c.

Below are comments from the first review round that had not been addressed yet because we did not receive the PDF with comments from the second reviewer. Those comments that we do not mention explicitly again below had already been addressed in the previous revision.

15) Reviewer comment:

The shape of DT cannot be seen by MRI? In some taxa of ascidians, DT shape is one of the key for species identification. Also you can mention the capability to apply MRI as non-destructive method in the discussion.

15) Response:

The dorsal tubercle shape is clearly visible in Figure 2a. It is also highlighted and measured in the MRI slice. We added the “non-destructiveness” of the MRI in the discussion (line 369).

16) Reviewer comment:

Mention and discuss about the differences in peaks among the regions (CS, MZ, PE) particularly the peaks of 3200–3500 exhibiting OH-bonds.

16) Response:

Thanks for pointing that out. The two peaks are not important to the question we asked in the FTIR. Nevertheless, we mentioned them to avoid confusion. We have now added the information regarding the C-H bonds and the O-H bonds, which represent two of the most prominent peaks in the FTIR graph.

17) Reviewer comment:

Could you attach a schematic figure of the ascidian indicating the position of the slices?

17) Response:

The slices are indicated in figure 2 parts a and c with dashed red lines. The dashed lines have small letters on the right that show the position of the corresponding figure parts 2a-c.